# In-host evolution of *Staphylococcus epidermidis* in a pacemaker-associated endocarditis resulting in increased antibiotic tolerance

Vanina Dengler Haunreiter[1], Mathilde Boumasmoud[1], Nicola Häffner[1], Dennis Wipfli[1], Nadja Leimer[1,6], Carole Rachmühl[1,7], Denise Kühnert[1,8], Yvonne Achermann[1], Reinhard Zbinden[2], Stefano Benussi[3], Clement Vulin[4,5] & Annelies S. Zinkernagel[1]

Treatment failure in biofilm-associated bacterial infections is an important healthcare issue. In vitro studies and mouse models suggest that bacteria enter a slow-growing/non-growing state that results in transient tolerance to antibiotics in the absence of a specific resistance mechanism. However, little clinical confirmation of antibiotic tolerant bacteria in patients exists. In this study we investigate a *Staphylococcus epidermidis* pacemaker-associated endocarditis, in a patient who developed a break-through bacteremia despite taking anti-biotics to which the *S. epidermidis* isolate is fully susceptible in vitro. Characterization of the clinical *S. epidermidis* isolates reveals in-host evolution over the 16-week infection period, resulting in increased antibiotic tolerance of the entire population due to a prolonged lag time until growth resumption and a reduced growth rate. Furthermore, we observe adaptation towards an increased biofilm formation capacity and genetic diversification of the *S. epidermidis* isolates within the patient.

[1] Division of Infectious Diseases and Hospital Epidemiology, University Hospital Zurich, University of Zurich, 8091, Zurich, Switzerland. [2] Institute of Medical Microbiology, University of Zurich, 8006, Zurich, Switzerland. [3] Department of Cardiac Surgery, University Heart Center, University Hospital Zurich, University of Zurich, 8091, Zurich, Switzerland. [4] Institute of Biogeochemistry and Pollutant Dynamics, ETH Zurich, 8092, Zurich, Switzerland. [5] Department of Environmental Microbiology, Eawag, 8600, Dübendorf, Switzerland. [6] Present address: Antimicrobial Discovery Center, Department of Biology, Northeastern University, 02115, Boston, MA, USA. [7] Present address: Institute of Food, Nutrition and Health, ETH Zurich, 8092, Zurich, Switzerland. [8] Present address: Max Planck Institute for the Science of Human History, 07745, Jena, Germany. These authors contributed equally: Mathilde Boumasmoud, Nicola Häffner. Correspondence and requests for materials should be addressed to A.S.Z. (email: annelies.zinkernagel@usz.ch)

Since the discovery of antibiotics, many bacterial infections have become treatable. However, antibiotic tolerance may limit treatment efficiency, resulting in chronic and relapsing infections, as found in biofilm-associated infections. Biofilms are sessile communities of bacteria that are attached to biotic or abiotic surfaces and are embedded in a matrix of extracellular polymeric substances[1]. The environment within a biofilm is heterogeneous, with nutrient limitation in the lower layers restricting growth. These non-growing or slow-growing bacteria are protected from antibiotics targeting active cell growth (reviewed in refs. [2,3]).

In contrast to antibiotic-resistant bacteria, tolerant bacteria remain fully susceptible to the antibiotic once they resume growth. While the antibiotic minimum inhibitory concentration (MIC) is the gold-standard metric to assess resistance, the minimum duration to killing (MDK) metric has been proposed to define tolerance[2]. The longer it takes to kill the bulk of a bacterial population, the more tolerant this population is. This time-span depends on the population's cell growth rate and for growth-arrested cells on their time to growth resumption. These growth parameters are determined by both the environment and the intrinsic properties of the strain. One can therefore distinguish phenotypic tolerance from genotypic tolerance[4]. On the one hand, the phenotypic tolerance is a transient state induced by a specific environment, such as low pH, nutrient limitation, or antibiotic challenge[5]. It often characterizes only a fraction of the population, referred to as persister cells. On the other hand, the genotypic tolerance involves mutations in the entire population. Upon antibiotic exposure, tolerance of a bacterial population was observed to evolve faster than resistance in vitro[6,7], and this initial adaptation has been described as an important step toward the development of antibiotic resistance[8,9].

Antibiotic tolerance has been studied in detail in vitro and in mouse models (reviewed in refs. [2,10]). The development of affordable whole-genome sequencing technologies allowed the analysis of in-host evolution in chronic infections, particularly of *Staphylococcus aureus*, *Escherichia coli*, *Enterococci* ssp., and *Pseudomonas aeruginosa* in cystic fibrosis patients' lungs (reviewed in refs. [11,12]). However, the adaptation toward increased antibiotic tolerance has not been investigated in detail within patients and has only been described for *Enterococcus*

*faecium*[13]. In the clinical setting, bacterial tolerance is still mainly restricted to the endpoint observations of antibiotic treatment failures caused by antibiotic-susceptible bacteria[9].

*Staphylococcus epidermidis* is one of the most frequent causes of medical implant-associated biofilm infections causing orthopedic-, pacemaker-, and prosthetic heart valve-associated infections[14]. Existing *S. epidermidis* studies using whole-genome sequencing analyzed bacterial transmission in hospitals and investigated typical hospital-associated clones from different sources[15–17]. To our knowledge, none focused on in-host evolution.

Here, we present in host evolution of a *S. epidermidis* strain repeatedly isolated from a patient with a pacemaker-associated endocarditis during an infection period of 16 weeks. The genotypic in host evolution of the *S. epidermidis* strains causing the infection results in distinct phenotypes. The clinical isolates obtained at later time points of the infection present increased biofilm formation, reduced growth rates, and prolonged times until growth resumption when tested in vitro, resulting in increased antibiotic tolerance.

## Results

**Clinical case.** An afebrile 39-year-old man was admitted to the University Hospital of Zurich due to a pacemaker pocket infection. This first pacemaker was implanted 22 years ago because of cardiac arrhythmia. A new pacemaker was then implanted on the contralateral side 14 years later because of lead dysfunction of the first pacemaker. The leads of the first inactive pacemaker were left in situ since they could not be removed without causing damage and thus were cut and capped. The batteries were replaced twice, 8 and 2 years prior to the infection.

Upon admission, the pocket of the inactive first pacemaker was debrided, and the electrodes were trimmed as they could not be completely removed without open-heart surgery. Intraoperatively, turbid fluid was found and sent for microbiological analysis. *S. epidermidis* grew in multiple tissue samples, and, accordingly, the empirical antibiotic treatment amoxicillin/clavulanate was changed to intravenous vancomycin and rifampicin (Fig. 1). Antimicrobial susceptibility testing revealed a methicillin-susceptible *S. epidermidis*, only resistant to ampicillin and erythromycin. Therefore, vancomycin treatment was

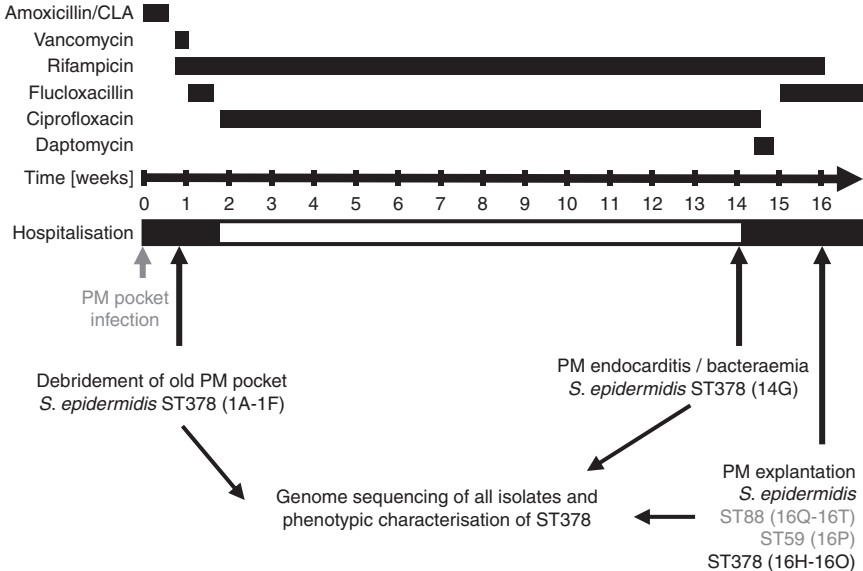

**Fig. 1** Overview of the clinical course, treatment, and *S. epidermidis* sampling. The antibiotic treatment scheme over the 16-week infection period. Surgical interventions and time points of *S. epidermidis* recovery are indicated. PM Pacemaker, ST sequence type

switched to flucloxacillin and rifampicin treatment was continued. The local infection significantly improved, and, after 12 days, antibiotic treatment was switched to an oral regimen consisting of rifampicin and ciprofloxacin and the patient was discharged from the hospital.

Fourteen weeks later, the patient presented with fever and was readmitted to the hospital. *S. epidermidis* grew in all four inoculated blood culture bottles, after 29 h under aerobic conditions and after 61 h under anaerobic conditions. A pacemaker-associated endocarditis was diagnosed after an echocardiography revealed two vegetations that were attached to the ventricle and the right atrial electrode, respectively. The empirically initiated systemic antibiotic treatment with daptomycin was changed to intravenous flucloxacillin, once methicillin susceptibility was confirmed and oral rifampicin was continued. Blood cultures did not show any bacterial growth 2 days later. Two weeks later, the pacemaker and all the leads were removed, and an epicardial pacemaker was implanted during an open-heart surgery. Despite the extensive antibiotic treatment, 70 *S. epidermidis* colony-forming units (CFUs) per ml were cultivated after sonication of the pacemaker aggregate and the leads. After completing the antibiotic treatment, the patient fully recovered and has been without an infection over the last 3 years as documented by clinical, laboratory, and echocardiographic follow-ups. An overview of the antibiotic treatment regimen and the isolation time points of *S. epidermidis* is shown in Fig. 1.

**Infection by susceptible strain under antibiotic treatment**. In this study, we characterized *S. epidermidis* clinical isolates obtained over a 16-week period from a pacemaker pocket infection progressing to a pacemaker-associated endocarditis. Isolates with different colony morphologies were detected. This resulted in six distinct bacterial isolates obtained from the pocket site infection at week 1 (isolates 1A–1F), one isolate obtained from the blood culture (isolate 14G) at week 14 and thirteen isolates from the explanted pacemaker aggregate and electrodes at week 16 (isolates 16H–16T, Table 1, Fig. 1). Multi-locus sequence typing (MLST) and pulsed-field gel electrophoresis (PFGE) revealed that the majority of isolates belonged to the same sequence type (ST) 378 and showed a very similar PFGE pattern

(Table 1, Supplementary Fig. 1). In addition to the ST378, five *S. epidermidis* isolates obtained from the patient belonged to other sequence types, four were ST88, and one was ST59. These other sequence types were only detected at one time point during the infection, at week 16. Using whole-genome sequencing, we aimed to investigate their relation to the ST378 isolates. The ST378 is a very rare sequence type, which so far has been isolated only once as a commensal in Sweden in 2008 (www.pubmlst.org). All ST378 isolates obtained from the patient showed the same resistance profile; resistance to ampicillin/penicillin and erythromycin, and susceptibility to all other antibiotics tested (including ciprofloxacin, see MICs in Supplementary Table 1). However, development of rifampicin resistance was observed in three out of the eight ST378 isolates obtained at week 16 (16M, 16K, and 16H, Table 1). The *S. epidermidis* isolate retrieved from the blood cultures at week 14 was fully susceptible to ciprofloxacin and rifampicin, the antibiotics that the patient was taking at that time. Thus, the break-through bacteremia under ciprofloxacin and rifampicin treatment indicated in vivo tolerance in the patient.

Since our focus was on the in-host evolution of the isolated *S. epidermidis* strains as well as the characterization of the observed in vivo tolerance of a susceptible *S. epidermidis* strain under antibiotic treatment in a patient, we next performed detailed phenotypic and genotypic analysis of the recovered ST378 isolates.

**Phylogenomics reveal a high mutation rate and two clusters**. All clinical *S. epidermidis* isolates recovered, including all sequence types (ST59, ST88, and ST378), were assessed by whole-genome sequencing. This revealed three completely independent strains that did not cluster together in a maximum-likelihood tree, including all complete genomes available for *S. epidermidis* on the National Center for Biotechnology Information database (www.ncbi.nlm.nih.gov, Fig. 2a, Supplementary Table 3). This confirmed that the two other sequence types (ST59 and ST88) recovered from the patient did not originate from the ST378 or vice versa.

Detailed analysis of the ST378 sequencing data analysis revealed an average total genome size of 2.44 Mb. We identified two plasmid replicons. The inferred plasmid 1 was present in all

**Table 1 List of *S. epidermidis* isolates obtained from the patient over the infection period**

| Isolate | ST | Phylogenetic group | Isolation week | Isolation material | Resistance profile |
|---------|-----|--------------------|----------------|--------------------|--------------------|
| 1A | 378 | 1 | Week 1 | Deep wound extract pocket site infection | PEN, AMP, ERY |
| 1B | 378 | 1 | Week 1 | Deep wound extract pocket site infection | PEN, AMP, ERY |
| 1C | 378 | 1 | Week 1 | Tissue pocket site infection | PEN, AMP, ERY |
| 1D | 378 | 1 | Week 1 | Tissue pocket site infection | PEN, AMP, ERY |
| 1E | 378 | 1 | Week 1 | Electrode (inactive PM) | PEN, AMP, ERY |
| 1F | 378 | 1 | Week 1 | Electrode (inactive PM) | PEN, AMP, ERY |
| 14G | 378 | 2 | Week 14 | Blood culture | PEN, AMP, ERY |
| 16H | 378 | 2 | Week 16 | Electrode, n.s. | PEN, AMP, ERY, RIF |
| 16I | 378 | 2 | Week 16 | Pacemaker aggregate | PEN, AMP, ERY |
| 16J | 378 | 1 | Week 16 | Pacemaker aggregate | PEN, AMP, ERY |
| 16K | 378 | 2 | Week 16 | Right ventricular electrode | PEN, AMP, ERY, RIF |
| 16L | 378 | 2 | Week 16 | Right atrial electrode | PEN, AMP, ERY |
| 16M | 378 | 2 | Week 16 | Right ventricular electrode | PEN, AMP, ERY, RIF |
| 16N | 378 | 1 | Week 16 | Silicon caps of electrodes (inactive PM) | PEN, AMP, ERY |
| 16O | 378 | 2 | Week 16 | Right atrial electrode | PEN, AMP, ERY |
| 16P | 59 | n.a. | Week 16 | Pacemaker aggregate | PEN, AMP, ERY |
| 16Q | 88 | n.a. | Week 16 | Silicon caps of electrodes (inactive PM) | PEN, AMP |
| 16R | 88 | n.a. | Week 16 | Electrode, n.s. | PEN, AMP |
| 16S | 88 | n.a. | Week 16 | Electrode, n.s. | PEN, AMP |
| 16T | 88 | n.a. | Week 16 | Electrode, n.s. | PEN, AMP |

*n.a.* not applicable, *n.s.* not specified, *PEN* penicillin, *AMP* ampicillin, *ERY* erythromycin, *RIF* rifampicin, *PM* pacemaker, *ST* sequence type

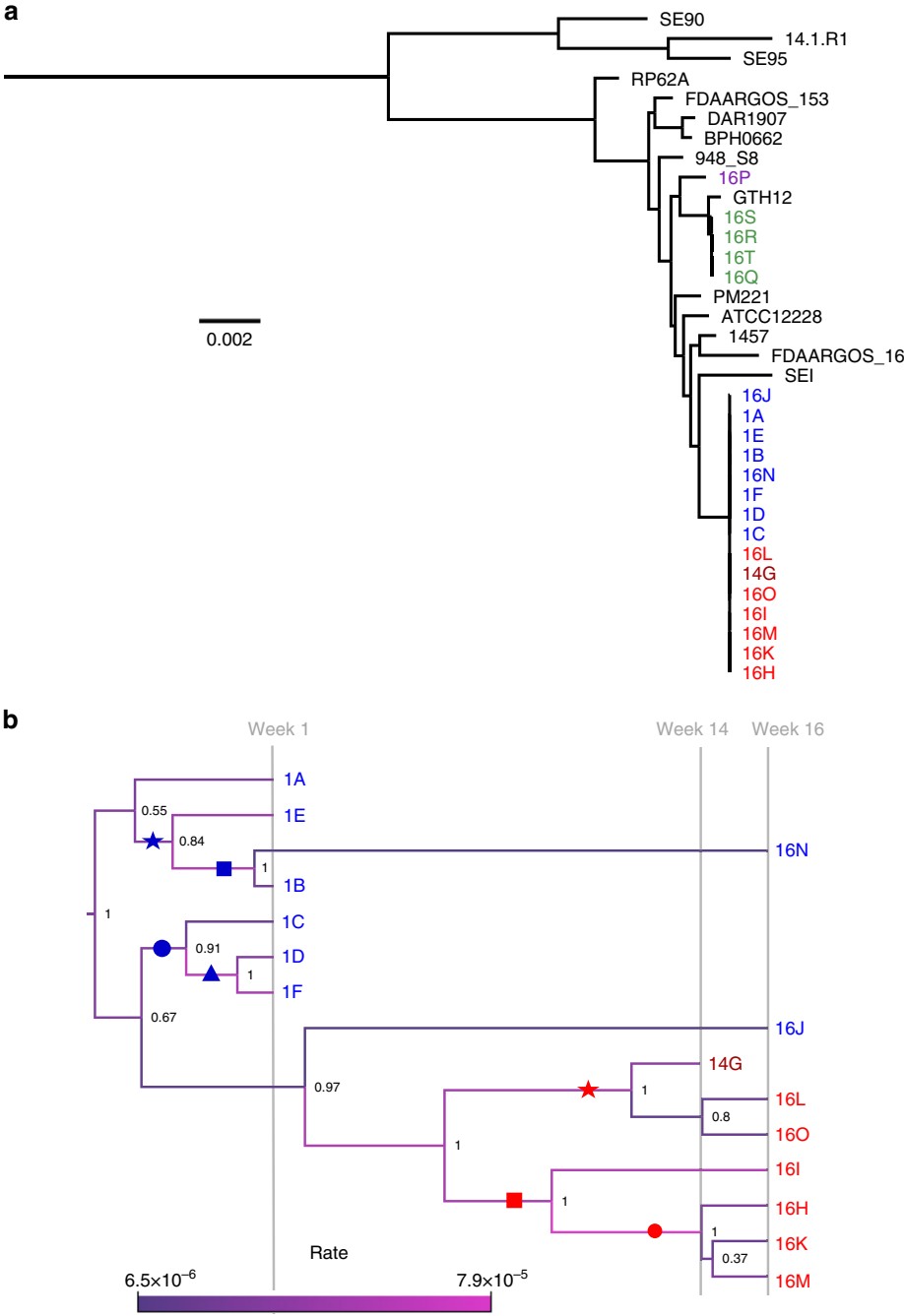

**Fig. 2** Phylogenetic trees of the *S. epidermidis* isolates. **a** Midpoint-rooted maximum-likelihood tree based on the core-genes alignment of all the clinical isolates (ST59 purple, ST88 green, ST378 blue/red), and all complete *S. epidermidis* reference genomes available on NCBI (black, Supplementary Table 2). The scale bar indicates 0.002 SNPs per site. **b** Bayesian Evolutionary Analysis Sampling Trees (BEAST) phylogeny based on the ST378 isolates core gene SNPs alignment. The branch length in this tree is determined by the sampling time point. The branch coloring illustrates the estimated mutation rate (SNPs per site per year), whose range is indicated by the scale bar. Group I and group II isolates are indicated in blue and red, respectively. Dark red indicates the clinical isolate retrieved from the blood culture. The symbols (blue star, square, circle and triangle and red star, square and circle) label subclusters within the two main clusters. Denotations of these symbols can be found in the last column of Tables 2 and 3, to relate the mutations to the subcluster they were found in

isolates, had an estimated size of 26 kb and encoded the beta-lactamase cassette *blaIR1Z* and the macrolide efflux pump *msr(A)* matching the phenotypically observed resistance to ampicillin/penicillin and erythromycin. Plasmid 2 was absent in two isolates, 1B and 16 N. It had an estimated size of 45 kb and encoded mainly hypothetical proteins as well as two toxin–antitoxin systems (YefM/YoeB and RelB/RelE). Moreover, in all ST378 isolates we identified a complete sequence of a STB20-like phage.

Investigation of known *S. epidermidis* virulence factors revealed the absence of two important factors for biofilm formation in all ST378 isolates recovered[18,19], the *ica* operon that encodes the biofilm polysaccharide intercellular adhesin (PIA) and *bhp* encoding the homolog of the biofilm-associated protein (Bap) of *S. aureus*. However, all strains contained the following genes relevant for biofilm formation, *sdrG, sdrH, embp, ebh, ebpS, aap, sbpS, fmt, atlE,* and *sle1*, indicating the potential of the strains to

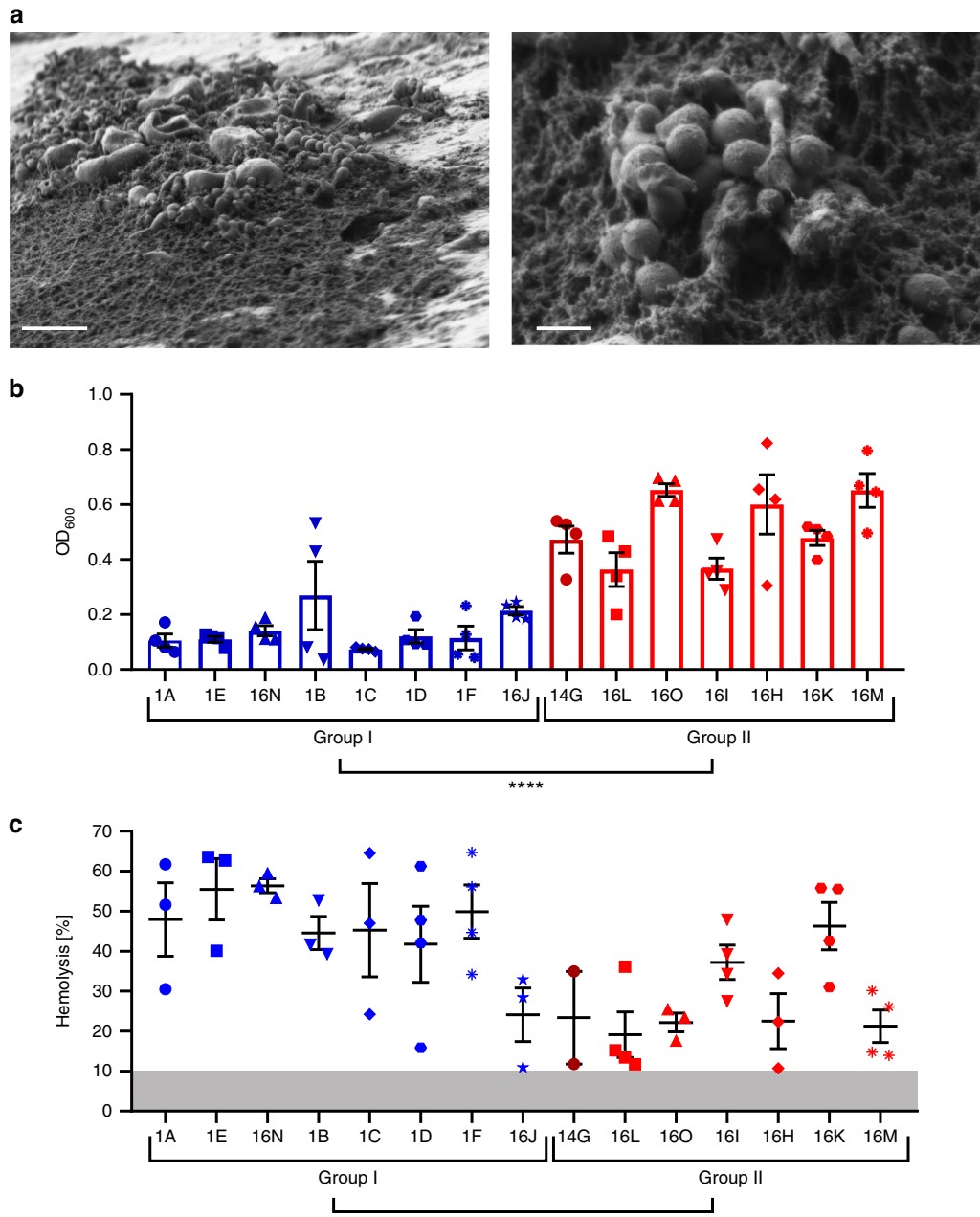

**Fig. 3** Biofilm formation and hemolysis of the clinical ST378 isolates. **a** FEM pictures of the biofilm formed on the patient's electrode. Scale bar left 5 µm and right 1 µm. **b** Quantitative in vitro biofilm assay of ST378 isolates. Isolates were grown in TSB 0.5% glucose in 96-well plates, and the washed biofilms were quantified by $OD_{600}$ measurement. Statistical significance between the two groups was determined by Welch's $t$ test ($N = 60$ $t(46.40) = -10.513$, $P = 7.3 \times 10^{-14}$). **c** Relative lysis of sheep blood erythrocytes by ST378 isolates indicated in percentages. The gray zone indicates the detection limit of 10% lysis below which the values were not included. Statistical significance between the two groups was determined by Welch's $t$ test ($N = 50$ $t(47.92) = 4.2857$, $P = 8.7 \times 10^{-5}$). Group I and group II isolates are indicated in blue and red, respectively. Dark red indicates the clinical isolate retrieved from the blood culture. Averages with standard error of mean of at least three replicates are shown. ****$P < 0.0001$

form biofilms, which we investigated in detail (Fig. 3). Regarding the presence of toxins, all strains encoded beta hemolysin (*hlb*) and a complete non-mutated set of phenol-soluble modulins (PMS, α, β, β1a β1b, β2, β3, δ, and ε), including delta toxin (Hld, PSM-γ). All isolates encoded a type III accessory gene regulator (*agr*) system. The insertion element IS256 frequently found in clinical *S. epidermidis* strains[20] was absent in all ST378 isolates. The gene content of the isolates regarding all known virulence factors[14] was generally identical, but the isolates differed in many single-nucleotide polymorphisms (SNPs) and short insertions or

deletions (InDels) potentially affecting virulence and biofilm formation as discussed below. Altogether, this highlights that the clinical ST378 isolate did not contain typical characteristics of hospital-associated *S. epidermidis* strains, which are frequently IS256, *SCCmec*, and *ica* positive and of *agr* type I and ST2[21,22].

A Bayesian Evolutionary Analysis Sampling Trees (BEAST) phylogeny based on the alignment of the SNPs in the core genes of the ST378 isolates revealed a sub-cluster of week 16 isolates (16 HIKLMO) and the bloodstream isolate (14G), which we named group II (labeled in red, Fig. 2b). Week 1 isolates (1A–1E) plus

week 16 isolates 16N and 16J were referred to as group I (labeled in blue, Fig. 2b). The isolate 16J falls between the two clusters, but closer related to the group I isolates as visible in the maximum-likelihood tree (Supplementary Fig. 2). The posterior node probabilities indicated considerable uncertainty at the branching event that gave rise to sample 1A (posterior probability 0.55) as well as the placement of sample 16J (posterior probability 0.67). However, the topology was robust under different model assumptions in the Bayesian analysis and agreed with the maximum-likelihood tree (Supplementary Fig. 2). Both week 16 isolates that clustered with the week 1 isolates, 16N and 16J, showed a reduced estimated mutation rate of about $7 \times 10^{-6}$ substitutions per site per year as compared with a more than 10-fold higher mutation rate of about $3 \times 10^{-5}$ for all other isolates (Fig. 2b, branch coloring). The mutation rate was generally high, but fell into the range of short-term evolutionary rate of $10^{-7}$–$10^{-5}$ substitutions per site per year observed for other bacteria during in-host evolution[12]. Isolates differed in pairwise comparison in up to 34 SNPs, indicating diversification of the *S. epidermidis* strain, and differences even increased when InDels were included (see below).

**Mutations in regulatory and metabolic genes**. As observed previously in in-host adaptation studies, we found many mutations in regulatory and metabolic genes. A complete list of all the 60 non-synonymous mutations (SNPs and InDels) differing between the isolates is given in the Tables 2 and 3. Table 2 shows the mutations characteristic of either all isolates, subclusters, or single isolates from the group I phylogenetic cluster. Table 3 shows the mutations characteristic of either all isolates, sub-clusters, or single isolates from the group II phylogenetic cluster. Subclusters are labeled with colored symbols on the phylogenetic tree (Fig. 2b) and denoted in the last column ("Phylogenetic cluster") in Tables 2 and 3. For example, the three rifampicin-resistant isolates (16 H, 16 K, and 16 M) belong to a subcluster within group II, labeled with a red circle on the phylogenetic tree. The mutations characteristic of this subcluster bear the "red circle" denotation in the "Phylogenetic cluster" column. An extended version of Tables 2 and 3, allowing a color-guided visualization and showing the mutations localization in the *S. epidermidis* RP62A reference genome can be found in the Supplementary Information (Supplementary Table 3).

Commonly observed host adaptations of the close relative *S. aureus* promoting persisting infections are mutations in the global regulators (*agr, sarA*) and the alternative sigma factor B likely resulting in decreased virulence, as well as in genes associated with the stringent response[12,23–26]. For the clinical *S. epidermidis* isolates recovered in this study, we observed mutations in all of those genes, including two *agrA* mutants, a *sarA* mutant, a mutation in *rsbU*, (a positive regulator of the alternative sigma factor B), and mutations in the stringent response genes *relQ, rsh*, and *codY*. This confirms similarities in in-host adaptation toward reduced virulence between *S. aureus* and *S. epidermidis*, even though *S. epidermidis* is known to have a much lower virulence potential[14].

Furthermore, we found evidence of a general selection pressure within the host since we observed multiple independent mutations in the genes *mqo, nrdl*, and *rpoB* (Tables 2 and 3). The latter, encoding the beta subunit of the RNA polymerase (RNAP), is well known to evolve upon rifampicin exposure[27]. We identified five independent mutations in *rpoB*, but only one was detected in all three rifampicin-resistant strains, namely an alanine insertion (Ala473_Asn474insAla). The other mutations were found in the rifampicin-susceptible strains (Supplementary Fig. 3, Supplementary Table 1).

In the following, we analyzed different phenotypes relevant for the pacemaker-associated biofilm infection to identify direct phenotypic consequences of the in-host evolution.

**In host adaptation toward increased biofilm formation**. The presence of a biofilm was confirmed on the explanted pacemaker electrode from the patient by field emission scanning electron microscopy (FEM) (Fig. 3a). Staphylococci embedded in an extracellular matrix were visible together with some host cells. In a next step, we analyzed the capacity of the different clinical isolates to form biofilm in vitro. Isolates from week 14 and 16 formed more robust biofilms under in vitro conditions (Fig. 3b). Isolates 16J and 16N showed less biofilm formation and were comparable with week 1 isolates (Fig. 3b), as reflected by the phylogeny (group I, Fig. 2b, Supplementary Fig. 2). The two *agrA* mutants, 16H and 16M, formed the thickest biofilm. However, the biofilms formed by the *agr* mutants were more susceptible to proteinase K and DNase I treatment as compared with the other group II isolates (Supplementary Fig. 4).

As a surrogate marker for toxin production reflecting virulence, we measured lysis of sheep blood erythrocytes (Fig. 3c) and detected an adaptation toward decreased toxin production in isolates recovered later during the infection period. Group I isolates showed significantly more hemolysis as compared with group II isolates: on average (± standard deviation) 46 (± 15)% and 28 (±13.5)%, respectively. Isolates 16J and 16K were outliers from their respective groups.

**Differences in growth characteristics**. We analyzed the growth characteristics of the isolates in the liquid medium (Fig. 4a). The growth curve indicated that group I isolates reached a certain $OD_{600}$, e.g., an $OD_{600}$ of 0.1, significantly faster than group II isolates (Fig. 4a, Supplementary Fig. 5a). This observation was due to the delayed growth in a sub-cluster of three isolates within group II (14G, 16L, and 16O). The minimal doubling time did not significantly differ between the two bacterial isolate groups (Supplementary Fig. 5b).

The different growth dynamics of the isolates were also reflected by their colony size at 24 h on agar plates (Fig. 4b). Group II isolates showed on average a significant smaller colony size as compared with group I isolates. The three isolates within group II, which had the slowest growth in liquid, also clustered in their colony size. They showed the smallest colony sizes among the group II isolates. These isolates (14G, 16L, and 16O) formed a monophyletic cluster in the phylogeny (Fig. 2b, denoted by a red star).

**Prolonged lag time and slower growth**. We and others reported previously that a lag time can cause differences in colony size[5,28]. To explore whether an increased lag time until growth resumption was contributing to a reduced colony size, we assessed colony growth kinetics of bacteria grown to stationary phase or under biofilm conditions in more detail. Three clinical isolates recovered at the three different time points during the infection (1A, 14G, and 16L) and reflecting the different colony sizes were assessed.

We observed a significantly higher radial colony growth rate for isolate 1A as compared with the isolates 14G and 16L (Fig. 5a, b, insets). Thus, the variation in colony size is partly explained by an altered growth rate. The week 14 and 16 isolates' (14G and 16L) colony growth curves showed a shift in time when compared with week 1 isolate (1A) (Fig. 5a, b). This suggested that a lag time contributed to the difference in colony size. To confirm that this observation reflected the growth dynamics at the microscopic level, we monitored the time to single cells' first division of stationary phase bacteria by time-lapse microscopy. We observed

**Table 2 Non-synonymous SNPs and InDels found in the group I clinical ST378 *S. epidermidis* isolates**

| ID # | Gene name/function | Amino acid change | Isolates | Phylogenetic cluster |
|------|--------------------|-------------------|----------|----------------------|
| 1 | GraR, two-component response regulator | Gly59Arg | 1A, 1B, 1C, 1D, 1E, 1F, 16J, 16N | Group I |
| 2 | Sodium/di- and tricarboxylate cotransporter | Ser161stop | 1A, 1B, 1C, 1D, 1E, 1F, 16J, 16N | Group I |
| 3 | Cold-shock protein CspA | Gly57fs | 1A, 1B, 1C, 1D, 1E, 1F, 16J, 16N | Group I |
| 4 | RodA, rod shape-determining protein/FtsW, cell division protein | Gly161Val | 1A, 1B, 1C, 1D, 1E, 1F, 16J, 16N | Group I |
| 5 | PbuG, hypoxanthine/guanine permease | Ser27Leu | 1A, 1B, 1C, 1D, 1E, 1F, 16J, 16N | Group I |
| 6 | RpoB, DNA-directed RNA polymerase beta subunit | Arg917Leu | 1A, 1B, 1C, 1D, 1E, 1F, 16N | |
| 7 | Sle1, autolysin, N-acetylmuramoyl-L-alanine amidase | Val67Ala | 1A | |
| 8 | NrdI, ribonucleotide reduction protein | Arg14stop | 1A | |
| 9 | SecDF, protein translocase subunit | Ser621Asn | 1A | |
| 10 | Bicyclomycin-resistance protein TcaB/major myo-inositol transporter IolT | Pro64Ala | 1A | |
| 11 | ClpC, ATP-dependent Clp protease ATP-binding subunit | Arg12_Gln18del | 1B, 1E, 16N | Blue star |
| 12 | NrdI, ribonucleotide reduction protein | Glu112fs | 1B, 1E, 16N | Blue star |
| 13 | Isocitrate dehydrogenase | Val356Leu | 1B, 1E, 16N | Blue star |
| 14 | SufB, Fe-S cluster assembly protein | Ala129Gly | 1E | |
| 15 | peptidase, U32 family large subunit [C1] | Glu90Gln | 1E | |
| 17 | PrmA, ribosomal protein L11 methyltransferase | Asp107Gly | 1B, 16N | Blue square |
| 18 | Sialic acid utilization regulator, RpiR family/MurR/RpiR family transcriptional regulator | Val84Ile | 1B, 16N | Blue square |
| 19 | Two-component sensor kinase WalK | Met428Thr | 1B, 16N | Blue square |
| 20 | SarA, Staphylococcal accessory regulator A | Ala70Thr | 16N | |
| 21 | 3′-to-5′ oligoribonuclease A | Pro46fs | 16N | |
| 22 | acetate kinase | Thr239Ala | 16N | |
| 23 | Mqo, malate:quinone oxidoreductase | Ser431_Pro432_Gly433_Ala434del | 16N | |
| 24 | Salicylate hydroxylase | Ala290Gly | 1C, 1D, 1F | Blue circle |
| 25 | Manganese ABC transporter, inner membrane permease protein SitD | Tyr243fs | 1C | |
| 26 | NrdI, ribonucleotide reduction protein | Gln54stop | 1C | |
| 27 | YrrC, RecD-like DNA helicase, deoxyribonuclease | Ile122Thr | 1C | |
| 28 | RpoC, DNA-directed RNA polymerase beta′ subunit | Gly433Val | 1D, 1F | Blue triangle |
| 29 | RibU, riboflavin transporter | Met1Val | 1D, 1F | Blue triangle |
| 30 | Iron-sulfur cluster assembly scaffold protein NifU | Lys134Glu | 1D | |
| 31 | PurR, pur operon repressor | Phe33Ser | 1F | |
| 32 | SrrB, respiratory response protein | Ile13Ser | 1F | |
| 33 | RpoB, DNA-directed RNA polymerase beta subunit | Gln137Pro | 16J | |
| 34 | Predicted RNA-binding protein, associated with RNAse of E/G family | Val15fs | 16J | |
| 35 | general stress protein 13 (contains ribosomal protein S1 (RPS1) domain) | Thr31fs | 16J | |
| 36 | Mqo, malate:quinone oxidoreductase | Arg375Trp | 16J | |
| 37 | PutP, proline/sodium symporter | Leu353stop | 1A, 1B, 1C, 1D, 1E, 1F, 16J, 16N, 14G, 16L, 16O | |

Non-synonymous mutations characteristic of all isolates, subclusters or single isolates from the group I phylogenetic cluster (labeled in blue in Fig. 2). The numbering in the first column is an identification number for each of the mutations reported (starting from 1 in Table 2, going up to 60 in Table 3). The isolates in which the mutations were found are given in the column "Isolates" and the symbols denotations in the "Phylogenetic cluster" column refer to the colored symbols shown on the phylogenetic tree branches to characterize subclusters (Fig. 2b). An extended version of this table, allowing a color-guided visualization and showing the mutations' localization in the *S. epidermidis* RP62A reference genome can be found in the Supplementary Information (Supplementary Table 3)

that the time for 80% of the population to resume growth differed by > 1.5 h between isolate 1A and isolates 14G and 16L (averages ± standard deviations: 2.3 ± 0.3 h, 4 ± 1.3 h, 3.8 ± 0.3 h, respectively) (Fig. 5c).

**In-host evolution resulted in antibiotic tolerance**. To assess a potential effect on antibiotic tolerance by the reduced growth rate and increased lag time, we investigated killing efficiency by highciprofloxacin concentrations. Ciprofloxacin time-kill curves confirmed an increase in bacterial survival for the late isolates 14G and 16L as compared with the early isolate 1A (Fig. 6b). After 3 h of highdose ciprofloxacin exposure, 69% and 65% of the late isolates 14G and 16L survived, respectively, as compared with 11% for the early isolate 1A. This means that the minimal time to kill 90% of the population ($MDK_{90}$) was 3 h for the early isolate 1A. Whereas $MDK_{90}$ was between 3 and 6 h for the clinical isolates 14G and 16 L. At 6 h, 7% and 5% of the population was still alive, as compared with <1% for the early isolate 1A. To determine whether this difference in survival after 3 h ciprofloxacin exposure reflected a global difference between group I as compared with group II, we tested the survival of all 15 isolates after

**Table 3 Non-synonymous SNPs and InDels found in the group II clinical ST378 *S. epidermidis* isolates**

| ID # | Gene name/function | Amino acid change | Isolates | Phylogenetic cluster |
|------|--------------------|-------------------|----------|----------------------|
| 38 | Pta, phosphate acetyltransferase | Asp245Tyr | 14G, 16H,16I, 16K, 16L, 16M,16O | Group II |
| 39 | Penicillin-binding protein 1/Cell division protein FtsI | Val605Leu | 14G, 16H,16I, 16K, 16L, 16M,16O | Group II |
| 40 | RsbU, sigma factor B regulator | Thr325Asn | 14G, 16H,16I, 16K, 16L, 16M,16O | Group II |
| 41 | RNA-binding protein, conserved protein domain family EVE RNA binding | Trp17Cys | 14G, 16H,16I, 16K, 16L, 16M,16O | Group II |
| 42 | ArlR, two-component response regulator | Leu51fs | 14G, 16H,16I, 16K, 16L, 16M,16O | Group II |
| 43 | Mqo, malate:quinone oxidoreductase | Arg132stop | 14G, 16H,16I, 16K, 16L, 16M,16O | Group II |
| 44 | RpoB, DNA-directed RNA polymerase beta subunit | Gly491_Gly492 insProGly | 14G, 16L,16O | Red star |
| 45 | MsrC, free methionine-(R)-sulfoxide reductase | Gly64Asp | 14G, 16L,16O | Red star |
| 46 | Sat, sulfate adenylyltransferase | Pro348Gln | 14G, 16L,16O | Red star |
| 47 | pyruvate dehydrogenase subunit beta/branched-chain alpha-keto acid dehydrogenase E1 | Glu201Lys | 14G, 16L,16O | Red star |
| 48 | MprF virulence factor, phosphatidylglycerol lysyltransferase | Glu791_His792 insLeuGlu | 16L | |
| 49 | GTP-sensing transcriptional pleiotropic repressor CodY | Glu254fs | 16H, 16I, 16K, 16M | Red square |
| 50 | glycerol-3-phosphate responsive antiterminator | Gly129Ala | 16H, 16I, 16K, 16M | Red square |
| 51 | PrmA, ribosomal protein L11 methyltransferase | Glu238Ala | 16H, 16I, 16K, 16M | Red square |
| 52 | RpoB, DNA-directed RNA polymerase beta subunit | Gly492Val | 16I | |
| 53 | RelQ, (p)ppGpp synthetase | Phe40fs | 16I | |
| 54 | RSH, (p)ppGpp synthase/hydrolase | Leu533Phe | 16H, 16K, 16M | Red circle |
| 55 | RpoB, DNA-directed RNA polymerase beta subunit | Ala473_Asn474 insAla | 16H, 16K, 16M | Red circle |
| 56 | HssR, heme response regulator | Thr95Ala | 16H, 16K, 16M | Red circle |
| 57 | AgrA, accessory gene regulator protein A | Glu42fs | 16H | |
| 58 | Hypothetical serine protease | Arg78Ile | 16H | |
| 59 | AgrA, accessory gene regulator protein A | Arg218Pro | 16M | |
| 60 | L-Cystine ABC transporter, periplasmic cystine-binding protein TcyA | Thr45_Tyr46del | 16M | |

Non-synonymous mutations characteristic of all isolates, subclusters or single isolates from the group II phylogenetic cluster (labeled in red in Fig. 2). The numbering in the first column is an identification number for each of the mutations reported (starting from 1 in Table 2, going up to 60 in Table 3). The isolates in which the mutations were found are given in the column "Isolates" and the symbols denotations in the "Phylogenetic cluster" column refer to the colored symbols shown on the phylogenetic tree branches to characterize subclusters (Fig. 2b). An extended version of this table, allowing a color-guided visualization and showing the mutations' localization in the *S. epidermidis* RP62A reference genome, can be found in the Supplementary Information (Supplementary Table 3)

ciprofloxacin challenge. We confirmed a significant difference between group I and group II, with on average (± standard deviation) 15 (±11)% and 61 (±19)%, respectively, of the bacterial population surviving (Fig. 6c).

After 24 h, less than 0.01% of planktonic bacteria survived ciprofloxacin treatment for all three strains (Fig. 6b, e, Supplementary Fig. 6a). However, the survival under ciprofloxacin drastically increased to 10–20% if bacteria were embedded in a biofilm (Fig. 6e). Survival of bacteria in pre-grown biofilms was only slightly further reduced to 5% when treated with a combination of rifampicin and ciprofloxacin. There was no difference observed between the early isolate and the late isolates in the overall proportion of bacteria killed by ciprofloxacin and the ciprofloxacin/rifampicin combination (Fig. 6e).

In addition to bacterial killing, we assessed the effect of antibiotics on biofilm integrity. Measuring the biomass by optical density, we found more biofilm remaining for 14G and 16L as compared with 1A (Fig. 6f).

**Discussion**

In this study, we show in-host evolution of a ST378 *S. epidermidis* strain, of which multiple spatially and temporally distinct isolates were recovered from a patient with a biofilm-associated pacemaker endocarditis. In addition to the ST378 strain, we isolated two other *S. epidermidis* strains of ST59 and ST88 from the same patient, suggesting a polyclonal infection. Polyclonal infections are not uncommon for *S. epidermidis* infections and were previously observed[17,29]. We cannot completely rule out that these two other sequence types were contaminants of the isolation

process, as *S. epidermidis* is a common commensal of the human skin and therefore prone to false positive detections.

We investigated in-host evolution of the ST378 isolates recovered from three different time points during infection, which only differed in a maximum of 34 SNPs. We performed in vitro assays to assess phenotypic characteristics in order to explore a link between the genetic mutations and phenotypes observed. We found that most of the isolates recovered later during the infection showed increased biofilm formation, decreased hemolysis, and an increased antibiotic tolerance, reflected by their higher MDK$_{90}$, as compared with the earlier isolates.

Since antibiotic tolerance depends highly on a strain's growth characteristics, we monitored bacterial growth both at the macroscopic and microscopic level. We found a significantly slower and delayed growth of the two late isolates 14G and 16L as compared with the early isolate 1A, suggesting that both slower and delayed growth were contributing to the increased antibiotic tolerance phenotype of these isolates.

Considering the short time-span of 16 weeks, we observed a high genetic diversity between the different isolates, which made it difficult to narrow down the phenotypic changes to a single mutation. We can speculate that mutations in metabolic genes like *mqo* could have an effect on growth. This gene was under selection pressure within the host, reflected by different mutation sites along with multiple branches of the phylogenetic tree. Similarly, we report multiple mutations in *rpoB*, encoding the beta subunit of the RNA polymerase (RNAP), and the target of rifampicin. *RpoB* was previously shown to evolve quickly both

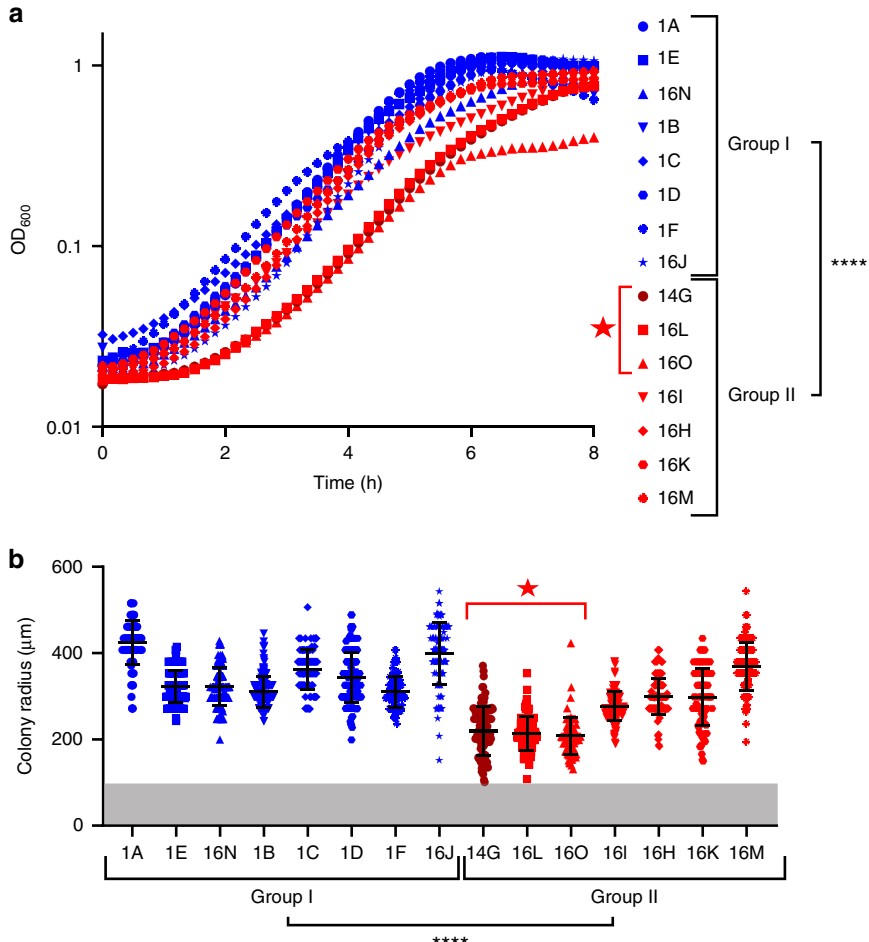

**Fig. 4** Growth characteristics of ST378 isolates. **a** Growth curves of *S. epidermidis* isolates in the liquid TSB medium. The mean growth curve of three replicates is shown. Statistical significance of the time to reach OD 0.1 between the two groups was determined by Welch's *t* test ($N = 46$ $t(29.96) = -5.0064$, $P = 2.295 \times 10^{-5}$). **b** Colony size for the different isolates after 24 h growth on sheep blood plates. Scatter dot plot with averages and standard deviations of 89–182 colonies per isolate are shown. The gray zone corresponds to a colony radius < 100 μm, which was the detection limit of our setup. Statistical significance between the two groups was determined by Welch's *t* test ($N = 1947$ $t(1587.8) = -19.634$, $P < 2 \times 10^{-16}$). Group I and group II isolates are indicated in blue and red, respectively. Dark red indicates the clinical isolate retrieved from the blood culture. The red star refers to the monophyletic cluster formed by the three isolates with delayed growth curves and smallest colonies at 24 h. ****$P < 0.0001$

in vitro and in vivo upon rifampicin exposure[27]. Mutations in specific positions of the coding sequence are known to affect the drug binding to the RNAP, leading to drug resistance. The specific insertion Ala473_Asn474insAla we detected in the three resistant strains and which we assume is responsible for rifampicin resistance has not been described so far. However, the same gene site was mutated (Ala473_Thr) in a rifampicin-resistant clinical *S. aureus* isolate[30].

While it is easy to deduce which mutation likely caused rifampicin resistance, determining which mutations might have affected tolerance is very difficult. Whether the mutation in RsbU, a positive regulator of the alternative sigma factor B, affects tolerance is difficult to judge by the findings of previous studies. Sigma B was shown to be crucial for SCV formation and persisting infections of *S. aureus*[23], but a *rsbU* mutation was shown to have no effect on antibiotic killing in another study[31]. Transposon mutagenesis studies further indicated that a limited number of single genes affect antibiotic tolerance, such as toxin–antitoxin systems and the stringent response pathway (reviewed in refs. [2,3]). However, the relevance of the stringent response in antibiotic tolerance is under debate for *S. aureus*, the closest relative of *S. epidermidis*[24,32].

It seems likely that tolerance is caused by a combination of different mutations. In this study, we observed mutations in genes affecting the stringent response pathway (*relQ*, *rsh*, and *codY*). However, two of the isolates we analyzed in more detail had a lag in growth resumption and did not show any mutation in the stringent response pathway. They had four mutations in common in the following genes: *rpoB*, the methionine-sulfoxide reductase *msrC*, the sulfate adenylyltransferase *sat* and the beta subunit of the pyruvate dehydrogenase (mutations 44–47, "red star" phylogenetic cluster, Table 3). It remains unclear which mutation or which combination was responsible for the increased lag time, which we confirmed in two different media (Fig. 5 and Supplementary Fig. 8), as these specific mutations have not been described before. The literature showed that mutations in *rpoB* can affect the growth rate of *S. epidermidis*[33], mutations in the other three genes have not been described yet in *S. epidermidis*. Thus, increased genotypic tolerance was likely due to both mutations in the stringent response pathway as well as mutations affecting the time to growth resumption. Mutant reconstruction to link specific mutations to a distinct phenotype could not be performed because the collected clinical *S. epidermidis* strains were refractory to genetic manipulation.

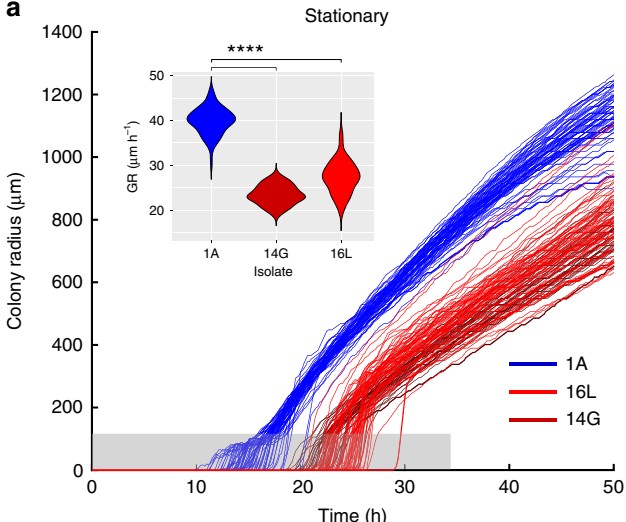

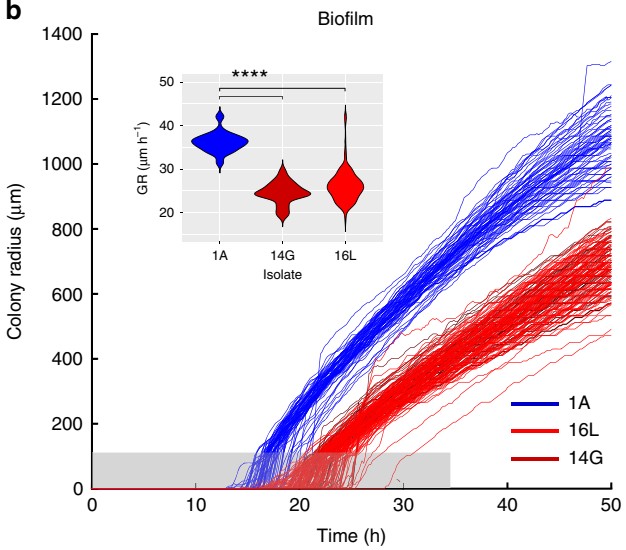

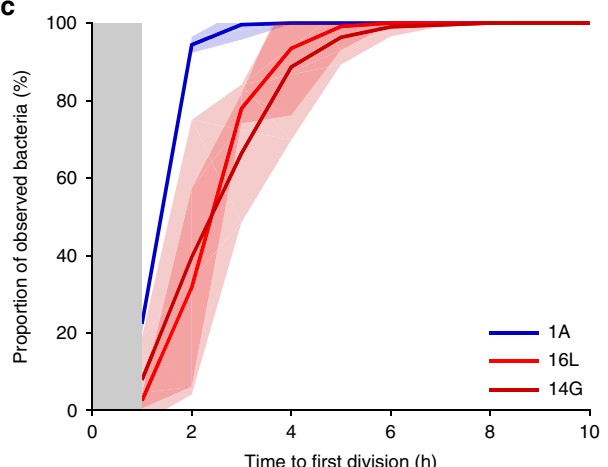

**Fig. 5** Macroscopic and microscopic analysis of the bacterial population's growth kinetics. **a** Colony growth curves of bacteria obtained from stationary phase cultures and **b** biofilms. The gray zone corresponds to a colony radius < 100 μm, which is under the detection limit of the macroscopic time-lapse setup. Small insert graphs show the distribution of the radial colony growth rate (GR) in μm h$^{-1}$ for the three isolates. Statistical significance between the three samples was determined by one way ANOVA (For the stationary: $N = 182$, $F(2,179) = 488.7$, $P < 2 \times 10^{-16}$, and Tukey's post hoc test $t_{14G-1A} = -16.45$, $p_{14G-1A} = 0$, $t_{16L-1A} -12.47$, $p_{16L-1A} = 0$/biofilm: $N = 237$, $F(2,234) = 311.8$, $P < 2 \times 10^{-16}$ and Tukey's post hoc test $t_{14G-1A} = -11.69$, $p_{14G-1A} = 0$, $t_{16L-1A} = -10.05$, $p_{16L-1A} = 0$). **c** Time to single cells' first division. Curves show averages of three replicates and shaded areas depict standard deviation. The gray zone marks the period at the beginning of the experiment where cell divisions could occur, but not be observed. Group I and group II isolates are indicated in blue and red, respectively. Dark red indicates the clinical isolate retrieved from the blood culture. ****$P < 0.0001$

observed in an *S. epidermidis* study[36]. However, this reduction of biofilm formation in a *S. epidermidis* ArlSR mutant observed by Wu et al. was *ica*-dependent; hence the *ica*-negative clinical isolates might not be affected. In addition to the mutations specific to the two groups of isolates also mutations in single isolates could affect their capacity to form biofilm. The two isolates of group II forming the thickest biofilm, 16H and 16M, contained *agrA* mutations (mutations 57 and 59, Table 3), potentially contributing to a more robust biofilm phenotype as described in previous studies. One study described increased biofilm formation by an isogenic *agr* mutant as compared with wild-type strain in a rabbit colonization model[37]. Another study reported that *RNAIII*, the gene encoding for the effector molecule of the *agr* system, was downregulated in clinical *S. epidermidis* isolates due to mutations in *agr*[38]. This repression resulted not only in increased biofilm formation but also in increased cell death and biofilm dispersal, which finally promoted new biofilm formation.

The mutation rate leading to this impressive diversification of a *S. epidermidis* strain within the medical implant associated biofilm was in the range of $10^{-5}$ substitutions per site per year, which is at the upper limit described for other bacterial species[12]. One study assessing transmission of *S. epidermidis* between patients in a hospital showed no genetic differences[39], which is in contrast to the diversity at a single time point observed here. However, a transmission bottle neck could be a potential explanation for the limited genetic variability.

Reconstructing the theoretical time point when the infection started, the BEAST analysis gave us an estimation of up to 4 weeks prior to the first sampling date, which coincides with the time when the patient presented with the erythema at the hospital. As the patient's last surgical intervention for battery replacement was more than 2 years prior to this infection, it seems unlikely that the pacemaker was contaminated at that time.

However, two isolates obtained at week 16 showed a reduced mutation frequency. Isolate 16N was obtained from the silicon caps of the wires of the inactive pacemaker located at the pocket site, the same location as all week 1 isolates with which it clustered in the phylogenetic tree. This suggests a potential niche adaptation. However, isolate 16J was obtained from the same agar plate of the sonicate of the active pacemaker aggregate as isolate 16I and the ST59 isolate. This highlights again the in-patient diversity as isolates 16J and 16I differed in 22 SNPs and InDels. Sequencing more colonies, including colonies that do not differ in morphology would have given a more complete picture of the infecting population and would have allowed a better interpretation of the in-host evolution. Still, based on our observation,

The increased biofilm formation by the isolates obtained later during the infection (group II) could have been due to a frameshift mutation in the response regulator ArlR. Previous studies showed that disruption of *arlSR* increased biofilm formation in *S. aureus*[34,35]. Contradictorily, reduction in biofilm formation was

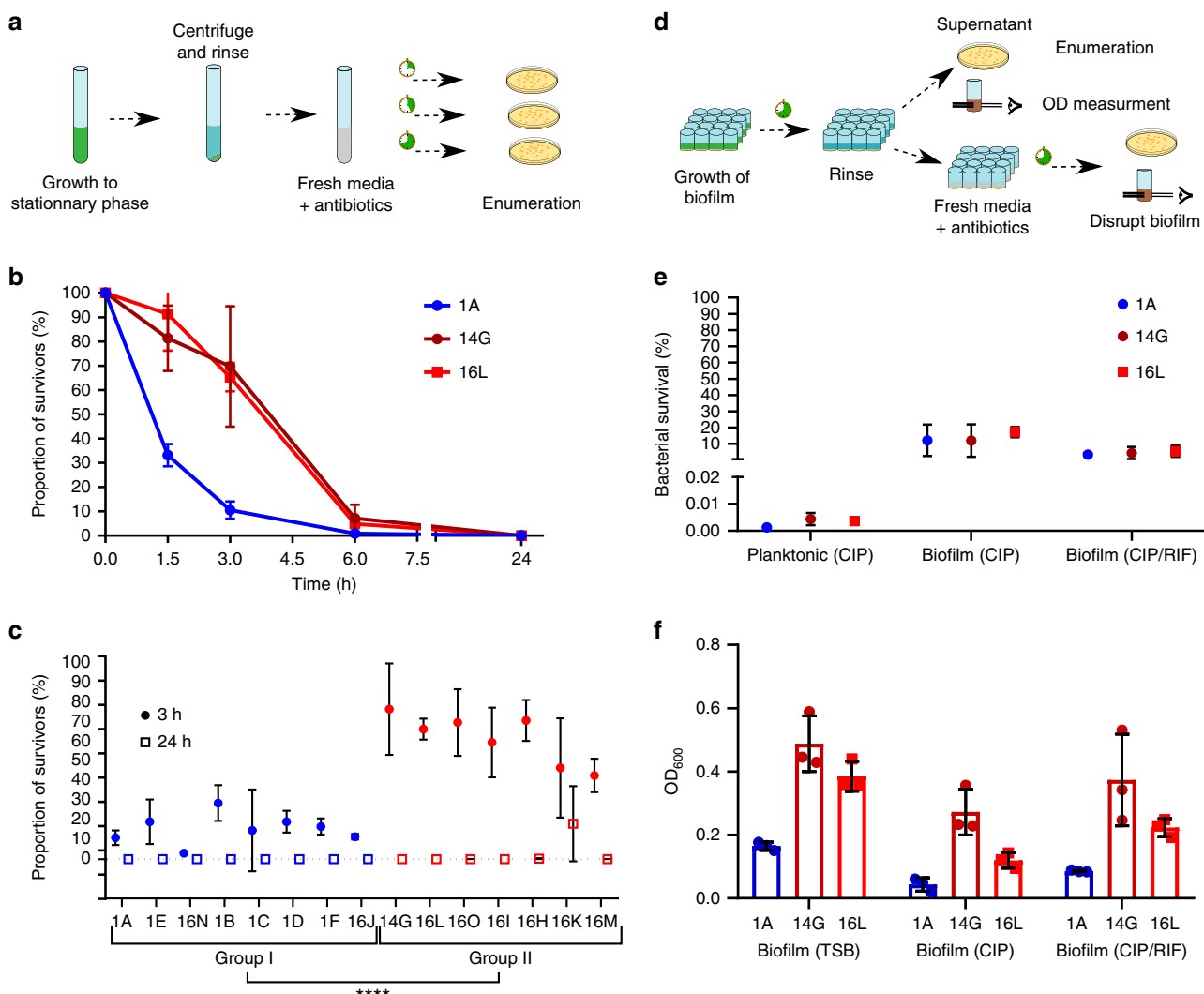

**Fig. 6** Antibiotic clearance of planktonically growing and biofilm-embedded bacteria. **a** Illustration of the assay used to determine the proportional killing of a bacteria population by antibiotics. **b**, **c** Ciprofloxacin killing of stationary growth phase bacteria. Bacteria were exposed to at least 40-fold MIC of ciprofloxacin for **b** 1.5, 3, 6, and 24 h and for **c** 3 and 24 h, respectively. The proportion of surviving bacteria relative to the inoculum was quantified by assessing the number of colony-forming units (CFUs) grown on agar plates. Curves and time points show averages of three replicates with standard deviation. Statistical significance of survival to 3 h antibiotic exposure between the two groups was determined by Welch's $t$ test ($N = 46$ $t(30.86) = -10.038$, $P = 3.069 \times 10^{-11}$). A log scale representation is given in Supplementary Fig. 6a, b. **d** Illustration of the assay used to determine the antibiotic treatment efficiency of biofilms. Bacterial killing within a biofilm was measured by enumerating CFUs, and disruption of the biofilm was quantified by optical density after treatment with antibiotics. **e** Antibiotic killing of bacteria embedded within a biofilm by ciprofloxacin and a combination of ciprofloxacin and rifampicin as compared with stationary phase grown bacteria (same data points as shown in **b**). The proportion of surviving bacteria relative to the bacteria recovered from the pre-grown biofilm was quantified by assessing the number of CFUs grown on agar plates. Averages with standard deviation of three replicates are shown. A log scale representation is given in Supplementary Fig. 6c. **f** Quantitative in vitro biofilm assays of untreated, ciprofloxacin, and a combination of ciprofloxacin- and rifampicin-treated biofilm. Biofilms were quantified by $OD_{600}$ measurement. Averages with standard error of mean of three replicates are shown. Group I and group II isolates are indicated in blue and red, respectively. Dark red indicates the clinical isolate retrieved from the blood culture. *****$P < 0.0001$

the cutoff values for the determination of transmission have to be chosen carefully if *S. epidermidis* biofilm-associated infections are involved.

The uneven mutation rate highlights that, in addition to their genetic background, the environment to which the isolates were exposed within the host affected their phenotypes. We hypothesize that the isolates with a lower mutation rate were in a slow-growing state within the biofilm, as compared with other isolates, for which the selection might have happened in the peripheral layers of the biofilm, where they were growing closer to a planktonic situation.

The delay in growth resumption and the lower growth rate we observed by time-lapse analyses were most likely crucial for the bacteria to survive within blood in presence of antibiotics and was reflected by the long time to positivity of the blood cultures (29–61 h). Furthermore, long time to positivity of the blood cultures typically reflects a low number of bacteria. The clearance of more than 99.99% of the bacteria under planktonic conditions within 24 h by ciprofloxacin, highlighted the relevance of the biofilm for long-term survival within the patient.

We hypothesized that the long-term survival of the bacteria was due to their localization within the biofilm. We confirmed

in vitro that the biofilm produced by this specific clinical *S. epidermidis* strain was not cleared by the antibiotics rifampicin and ciprofloxacin, which are often given in clinics to treat biofilm-associated infections and which the patient was given at the time of the break-through bacteremia.

Although we did not show a difference in survival, we observed a thicker biofilm remaining after antibiotic treatment in the two late isolates 14G and 16L as compared with the early isolate 1A. Dai et al. described increased extracellular biofilm matrix production in clinical *S. epidermidis* due to enhanced *atlE*-induced autolysis[38]. Furthermore, cell lysis was described as relevant source of the *ica*-independent biofilm matrix composed of eDNA and cytoplasmatic proteins in *S. aureus*[40,41]. The importance of cell lysis for biofilm formation might be an explanation why isolate 1A formed a weaker biofilm even though there were more viable bacteria within both the treated and untreated biofilm as compared to 14G and 16L isolates (Supplementary Fig. 7).

Altogether, we conclude that the bacteremia despite the presence of antibiotics was most likely a consequence of bacterial seeding from the biofilm and survival of *S. epidermidis* in the blood due to the observed tolerance phenotype.

In this study, we have focused on the genotypic basis of antibiotic tolerance. We observed an increase in lag time and decrease in growth rate that affected the entire population, but this does not rule out that subpopulations of persister cells contributed to antibiotic treatment failure[3,10]. We did observe a subpopulation of smaller colonies when bacteria were initially plated from the pocket site infection and from the explanted pacemaker. Those colonies could be an indication of phenotypic tolerance[5]. This heterogeneity of colony size within the same isolate was lost when the colonies were frozen and regrown.

To conclude, we showed for the first time in-host adaptation of *S. epidermidis*. We confirmed in vitro that the biofilm formed by the *S. epidermidis* isolates was not cleared by the antibiotics used in clinics. Even though a combination therapy consisting of ciprofloxacin and rifampicin was applied, rifampicin resistance evolved over time in this patient. We observed *S. epidermidis* bacteremia despite fully effective antibiotic treatment highlighting in vivo tolerance, which we confirmed with in vitro studies. This study confirms the relevance of antibiotic tolerance, so far only characterized in vitro, for chronic and difficult-to-treat bacterial infections as observed in this patient with an *S. epidermidis* medical-device-associated biofilm infection.

## Methods

**Ethical requirement**. Informed patient consent was obtained (Cantonal ethic commission Zurich).

**Bacterial strains and growth conditions**. Clinical *S. epidermidis* isolates (Table 1) were isolated by sonication from the pacemaker as described before[42] and characterized at the Institute of Medical Microbiology of the University of Zurich. For each clinical sample colonies with different morphology were archived if present, otherwise only one colony was frozen. Species identification was done by Microflex LT mass spectrometer (Bruker Daltonik) as described before[43].

The strains were stored in Lennox Broth (LB) supplemented with 20% glycerol at −80 °C. Bacteria were grown on solid agar plates containing Tryptic Soy Broth (TSB, BD) or columbia blood agar plates (BioMerieux) for ~40 h, if not indicated otherwise. Overnight cultures (o/n) were inoculated from fresh plates in 5 ml of TSB in 50 -ml conical tubes and incubated for 18–20 h under shaking conditions (220 rpm) at 37 °C. Bacterial growth was measured by optical density at 600 nm ($OD_{600}$) using a WPA CO8000 Cell Density Meter (Biochrom, Berlin, Germany) or a SpectraMax i3 multi-mode microplate reader (Molecular Devices, San José, CA, USA).

**Antibioticresistance evaluation**. Disc diffusion assays were performed by the Kirby–Bauer method according to the EUCAST guidelines using antibiotic disks (SirscanDiscs™, i2a, (Montpellier, France)) for ampicillin, amoxicillin/clavulanic acid, cefoxitin, vancomycin, teicoplanin, norfloxacin, ciprofloxacin, levofloxacin, moxifloxacin, amikacin, gentamicin, tobramycin, tetracycline, erythromycin, clindamycin, sulfamethoxazole/trimethoprim, and rifampicin on Mueller–Hinton

plates. EUCAST breakpoints were used to assign a strain to be resistant or susceptible according to the inhibition zone diameter.

MICs were determined using broth microdilution in the Mueller–Hinton medium according to EUCAST guidelines. MICs for ciprofloxacin (Bayer), erythromycin (Sigma), and rifampicin (Labatec) were determined.

**Field Emission Scanning Electron Microscopy**. Field emission scanning electon microscopy (FEM) was performed by the Imaging and Chemical Analysis Laboratory of the Montana State University using a Zeiss SUPRA 55VP. After explantation, one part of the electrode was immediately fixed in 2.5% glutaraldehyde in 0.1 M cacodylate buffer. The electrode was rinsed four times with distilled water for 15 min. Dehydration was performed with increasing concentration of ethanol, 20 min 25%, 20 min 50%, 20 min 75%, 30 min 95%, and three times 1 h in 100% ethanol prior to crucial point drying.

**Pulsed-field gel electrophoresis**. Pulsed-field gel electrophoresis (PFGE) typing of *SmaI*-digested genomic DNA was performed as recommended by Chung et al.[44]. Briefly, *S. epidermidis* o/n cultures were washed and resuspended in SE buffer containing 2% low melt agarose to form plugs. Plugs were incubated with lysozyme and mutanolysin in EC buffer overnight at 37 °C, with continous shaking. EC buffer was aspired and EC buffer containing proteinase K was added overnight at 50–56 °C, with continous shaking. Plugs were washed and equilibrated in $T_{10}E_{10}$ buffer. Bacterial DNA containing plugs were restricted with *SmaI*. Digested DNA was loaded on a 1% agarose gel and stained with ethidium bromide.

**Quantitative biofilm assays**. *S. epidermidis* o/n cultures were diluted to an $OD_{600}$ of 0.05 in 200 μl of the TSB medium supplemented with 0.5% glucose and grown statically at 37 °C for 24 h in 96-well microplate (Greiner Bio-One). Supernatants were discarded, biofilms on the well bottom were washed twice with 100 μl of PBS and resuspended in 200 μl of PBS. The biomass of the resuspended biofilm was determined by $OD_{600}$ measurement. To determine matrix composition, proteinase K (Omega Bio-Tek) at 0.1 mg/ml and DNase I (Roche) at 10 U/ml were added at inoculation, and biofilm formation was quantified after 24 h with crystal violet. Therefore, wells were washed three times with 200 μl of PBS, dried at room temperature before staining with 0.1% crystal violet for 30 min. After three wash steps with distilled water, adhering dye was dissolved with 30% acetic acid, and the absorption was measured at 570 nm. To determine the effect of antibiotics on biofilm integrity antibiotic media exceeding the MICs at least 40-fold were added onto the 24 h pre-grown biofilms (20 μg/ml ciprofloxacin or 20 μg/ml ciprofloxacin and 12 μg/ml rifampicin in TSB 0.5% glucose). Biomass was determined by $OD_{600}$ measurement of the resuspended biofilm.

**Growth curves**. *S. epidermidis* o/n cultures were diluted to an $OD_{600}$ of 0.02 in 200 μl of the TSB medium in a 96-well plate and incubated in a VersaMax microplate reader (Molecular Devices) for 20 h at 37 °C under constant shaking. The $OD_{600}$ was measured every 10 min. The minimal doubling time was calculated from the $OD_{600}$ reads using 1 -h intervals. Student's *t* test was used to determine the difference between the two groups and between two individual isolates.

**Antibiotic persister assays**. *S. epidermidis* o/n cultures were diluted to an $OD_{600}$ of 0.05 which corresponded to about $2 \times 10^7$ CFUs/ml in the antibiotic media (20 μg/ml ciprofloxacin), exceeding the MIC at least 40-fold. At inoculation and after 3 and 24 h incubation for all isolates and additionally 1.5 h and 6 h for isolates 1A, 14G, and 16L in the antibiotic medium, bacteria were washed twice with PBS before determination of CFUs by plating of serial dilutions.

**Antibiotic exposure of biofilms**. Biofilms were grown as described above for 24 h in TSB 0.5% glucose. Supernatants were discarded, and biofilms were washed twice with 100 μl of PBS before 200 μl of antibiotic media exceeding the MICs at least 40-fold were added onto the biofilms (20 μg/ml ciprofloxacin or 20 μg/ml ciprofloxacin and 12 μg/ml rifampicin in TSB 0.5% glucose). After 24 h incubation time, the antibiotic medium was removed, and the biofilms were washed twice and resuspended in 200 μl of PBS. CFUs were determined by plating of serial dilutions.

**Quantitative hemolysis assay**. Overnight cultures were grown in the Todd Hewitt (TH) medium as recommended by Quiblier et al.[45], adjusted to $OD_{600}$ 2, centrifuged, sterile filtered, and 100 μl were added to 100 μl of washed 5% sheep blood erythrocytes in PBS obtained from defibrinated sheep blood (Thermo Fisher). After incubation at 37 °C for 30 min and at 4 °C for 30 min, hemoglobin absorbance in the supernatant was measured at 415 nm.

**Automated agar plate imaging**. Overnight cultures or washed and resuspended biofilms were plated onto blood agar plates (Columbia + 5% sheep blood, Biomerieux) and placed in a 37 °C incubator. Images were taken by Canon EOS 1200D reflex cameras every 10 min for 48 h. Cameras were triggered by Arduino Uno board and optocouplers. Colonies' growth curves and radial growth were obtained by analyzing the images with an in-house software. The following number of

colonies were analyzed for stationary phase (1A: 75, 14G: 44, 16L: 66) and for biofilms (1A: 71, 14G: 65, 16L: 114).

**Single-cell time-lapse microscopy.** Bacteria were plated from frozen stock onto blood agar plates (Columbia + 5% sheep blood, Biomerieux). For inoculation, bacteria were harvested from the plate into TSB or Dulbecco's Modified Eagle Medium (DMEM) (4.5 g/l glucose, 10% FBS). The cultures were grown for 24 h and diluted to OD 0.1 in their respective media. Diluted bacteria were streaked onto blood agar pads containing Columbia media (BD) with 2% agar (bacteriological grade, BD) and 5% sheep blood (Life Technologies). Agar pads were covered with cover glasses and placed under the microscope at 37 °C. Bright field images were taken every 30 min no later than 30 min after initial inoculation at 100× (U-FLN-Oil lens) with an automated Olympus IX81 inverted microscope. Up to 3000 positions per experiment were recorded and the lag time of 90–189 bacteria was analyzed using the Cellsense software. Time to cells' first division was manually determined using ImageJ software[46].

**Whole-genome sequencing and assembly.** Total DNA of *S. epidermidis* clinical isolates was extracted from a single colony inoculated in the liquid medium and cultivated overnight using the DNeasy Blood & Tissue Kit (Qiagen), with an additional enzymatic lysis step using lysozyme and lysostaphin. DNA was quality checked with a Bioanalyzer (Agilent Technologies). Sequencing libraries were constructed with a Nextera® XT kit (Illumina) for 13 strains and with a NEBNext® Ultra™ kit (New England BioLabs) for isolates 16N and 16O, according to the manufacturers' recommendations. The sequencing was conducted on an Illumina MiSeq machine with a read-length of 2 × 150 bp (2 × 300 bp for isolates 16N and 16O). The quality of the sequencing data was evaluated using FastQC (v0.11.5, available at: http://www.bioinformatics.babraham.ac.uk/projects/fastqc/). CLC Genomic Workbench 10 (CLC, Qiagen) was used for adapter and quality trimming, resulting in 4'460'726 to 7'690'9632 reads for the Nextera XT library run. For the NEB Next Ultra, 472'430 and 503'006 reads were left after trimming with CLC for strains 16O and 16N, respectively. This corresponded to an approximated average coverage of the genome of *S. epidermidis* of at least 250-fold and 50-fold, respectively. De novo assemblies were generated using CLC and annotated by RAST[47], resulting in 51 to 72 contigs (average N50: 76'386.) for the isolates with 2 × 150 bp reads and in 38 and 42 contigs (N50: 125'473 and 133'023) for 16O and 16N, respectively.

**Genetic variants calling.** Detailed comparative analysis of single nucleotide polymorphisms (SNPs) and short insertion and deletion (InDels) was performed by mapping the reads to the reference genome of *S. epidermidis* RP62A (ATCC 35984, Supplementary Table 2) as well as by mapping the reads to the de novo assembly of isolate 1A with CLC default settings (minimal frequency of 90% and a minimal coverage of 10 reads). Non-specifically mapped reads were ignored. In all, 100% of the 1A de novo assembly was covered by all isolates, except for 1B and 16N which covered 98%. Mapping to the reference sequence of RP62A resulted in an average coverage of 88%. This combined approach allowed us to cover almost the whole sequence of the ST378 strains in the analysis, except for regions at the start and end of contigs with insufficient mapping quality. Furthermore, comparison with the reference allowed to dissect which variant corresponded to the reference and which was an alternative allele in our isolates. The 13'658 identical SNPs found in all isolates when compared with the reference (Supplementary Data 1) were subtracted for further comparative analysis. The mutations unique to one isolate or to a cluster of isolates are shown in Tables 2 and 3 (only non-synonymous mutations) and Supplementary Table 3 (all mutations).

**Pan-genome construction and phylogenetic analysis.** De novo assemblies as well as 14 published *S. epidermidis* assemblies (Supplementary Table 2) were annotated with Prokka[48]. The output was used to construct a pan-genome with Roary[49], using a 100% cutoff (core genes are shared by all strains) and a 95% default identity cutoff. This was done for all clinical isolates (ST59, ST88, and ST378) together with the reference strains yielding in 1'767 core genes concatenated into a single pseudo-sequence of 1'665'907 nucleotides (nt) and individually for the ST378 isolates only (2186 core genes, pseudo-sequence length 1'967'375 nt). The resulting alignments were used to build phylogenetic trees. The maximum-likelihood trees were created with FastTree[50] using the generalized time-reversible (GTR) model of nucleotide evolution. The Bayesian evolutionary analysis was performed in BEAST2[51] using an Hasegawa–Kishino–Yano (HKY) substitution model, a relaxed clock model with lognormally distributed branch rates and a birth–death-sampling tree prior[52]. The input was the alignment of the SNPs extracted from the ST378 strains core-genes alignment with the SNP-sites script (https://github.com/sanger-pathogens/snp-sites), gap-deleted and corrected for base compositional bias. Bayesian Markov-chain Monte Carlo (MCMC) convergence was confirmed in Tracer[53] and the summary tree was constructed using TreeAnnotator. All trees were visualized with FigTree[54].

**Identification of plasmid replicons and virulence factors.** The presence of plasmid replicons and virulence genes was investigated using the srst2 package, with the default 90% coverage and 10% divergence cutoff[55]. The plasmid Finder Database and the Virulence Factor Database (VFDB) were used, respectively.

**Identification of phages, resistance genes, and MLST.** Phages were analyzed using PHASTER[56]. Antibiotic resistance genes and multilocus sequence typing (MLST) were determined using the tools of the Center for Genomic Epidemiology RESfinder[57,58].

**Reporting summary.** Further information on experimental design is available in the Nature Research Reporting Summary linked to this article.

## Code availability
In-house software written for the time-lapse images analysis of this study is available at GitHub (https://github.com/clementvulin/SCVLongLag).

## Data availability
All raw sequence data have been submitted to the European Nucleotide Archive (ENA) under project "PRJEB27742". All the data produced for this study are available from the corresponding author upon request or at "Figshare".

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

## Acknowledgements

We thank our patient for allowing us to perform this study. The study was funded by the Swiss National Science Foundation grants 310030_146295 and 31003A_176252 to A.S.Z., by a Gottfried und Julia Bangerter-Rhyner-foundation grant to A.S.Z. and V.D.H., by a Swiss National Science Foundation Marie Heim-Vögtlin grant (PMPDP3_171320/1) and funding by the Max-Planck Society to D.K. and by a CASCADE-FELLOWS PCOFUND-GA-2012-600181 grant and funding by the ETH Zurich to C.V. This work was supported by the Clinical Research Priority Program of the University of Zurich for the CRPP "Precision medicine for bacterial infections".

## Author contributions

D.W., N.H., V.D.H. and C.V. performed phenotypic characterization. N.L. and C.R. collected the strains and performed initial characterization. V.D.H. and M.B. performed bioinformatic analysis of the sequencing data. M.B. and D.K. did phylogenetic analysis and tree construction. N.H., M.B., C.V. and D.W. analyzed the time-lapse data. Y.A., A.S.Z., R.Z. and S.B. were responsible for the clinical and microbiologic work-up. V.D.H., M.B., N.H. and A.S.Z. wrote the paper.

## Additional information

**Competing interests:** The authors declare no competing interests.

