## [Peer Review File · Nature Communications]

Reviewers' comments:

Reviewer #1 (Remarks to the Author):

REVIEW NCOMMS-18-22067

The paper entitled " In-host evolution of *Staphylococcus epidermidis* in a pacemaker associated endocarditis resulting in increased antibiotic tolerance" by Dengler Haunreiter et al. corresponds to a genetic and phenotypic study of *S. epidermidis* isolates originating from a pace maker associated endocarditis.

This type of study analyzing the evolution of isolates originating from the same patient is actually relatively rare and can bring important information on the evolutionary path(s) taken by bacteria when exposed to different stresses within the host. For this reason this work is original and actually represents a first demonstration of a clinically relevant in host evolution of *S. epidermidis*.

In this manuscript the authors remained cautious, may be because this study is performed only with a single patient. This makes it difficult to provide very general conclusions.

The authors did actually make one very strong claim. Their main conclusion is indicated in the last 3 lines of the discussion where they wrote that their study confirms the relevance of antibiotic tolerance for chronic and difficult-to-treat bacterial infections. Such demonstration could be very important and studies demonstrating the role of antibiotic tolerance in the failure of classical antibiotic treatment(s) to clear infection are scarce.

However, with the current version of this manuscript, I am not totally convinced by the demonstration of the tolerance character of the isolates (see below). I think that this study also suffers from the fact that no clear conclusions can be drawn on the mutations that are responsible for the observed phenotypes since no mutation reconstruction or reversion to the wild type genotype have been performed.

Major concerns:

- line 206-213: the authors indicated that only one mutation in *rpoB* observed in isolates 16H, K and M was responsible for rifampicin resistance. To me this can be claim only if the mutation is

created in a clean (wt) background and if the mutant displays rifampicin resistance. Other unique mutations are found in the isolated 16H, K and M that could be possibly be involved in rifampicin resistance (for example Rsh leu533phe). Additionally isolates 16H, K and M share some mutations with other isolates but it could be that this is the combination of some of this shared mutations with others present in 16H, K and M and absent in others that could be responsible for the rifampicin resistance. Is there any indication in the literature that this specific mutation, if found in other studies, is responsible for rifampicin resistance?

- It is difficult to follow the evolution paths leading to the different isolates. To help the reader to follow the evolution of the isolates it will be good to provide a table where for each isolate is indicated all the identified mutations. In addition, it would be very helpful to have a sort of phylogenetic representation of the isolates where the potential overtime emergence of the mutations is indicated, eventually together with the potential associated phenotypes and the sites of isolation of the isolates.

- Why the isolate 16N has not been characterized phenotypically?

- One of the important results of the manuscript is the apparent evolution of the isolates toward increase biofilm formation capacity. Again, it would be good to have a formal demonstration that the detected agrA mutations are actually responsible for this phenotype.

- Concerning the demonstration of the enhanced tolerance of late isolates: in Figure 5C and 5D it would be more accurate to represent the bacterial survival as log of the % of bacterial survival. This representation generally reveals more easily important tolerance differences between isolates. No statistics are provided here to support the hypothesis of enhanced tolerance of the late isolates. Since this is one of the main message of the manuscript it would also be important to potentially show that what is shown for one early and two late isolates also applied to the different early and late isolates from this study. Brauner et al in their 2017 manuscript (Brauner, A., Shoresh, N., Fridman, O., & Balaban, N. Q. (2017). An Experimental Framework for Quantifying Bacterial Tolerance. *Biophysical Journal*, 112(12), 2664–2671. <http://doi.org/10.1016/j.bpj.2017.05.014>) proposed an elegant way of measuring/demonstrating enhanced tolerance using the calculation of the MDK parameter. The measurement of such as parameter could strengthened the results of this manuscript. One main question that remains unanswered here is what mutation(s) potentially explain the enhanced tolerance of late isolates during the planktonic phase? By using all the isolates in this tolerance assay it could be easier to narrow down common mutations present in all the tolerant isolates. This could eventually for a part replace the necessity to reconstruct identified mutation(s) to validate their association to the tolerance phenotype. In Figure 5C the authors showed that biofilms from early or late isolates did not display difference in tolerance. Does this mean that the mutations causing enhanced tolerance in planktonic do not contribute to tolerance of biofilms? Since the main lifestyle of bacteria when growing in a prosthetic device is the biofilm lifestyle is it that the enhanced tolerance observed in planktonic situation does actually not contribute to in host tolerance? Why these mutations have been somehow selected if at the end they did not really contributed to the tolerance phenotype in the host? It seems to me that the authors should clarify here their message regarding tolerance in host.

Minor concerns:

- one isolate belongs to the ST59 type (isolate 16P). Why no phenotypic analysis has been done on this isolate?
- Isolate 16J from group I follows most of the phenotype of group I with the exception of hemolysis phenotype that resemble group II (figure 3C). Any mutation(s) that could explain this phenotype?

Typo errors:

- line 277: Thus, the variation in in the colony growth ... There is a "in" to remove.

Reviewer #2 (Remarks to the Author):

Antibiotic tolerance is an important clinical problem that is very rarely pursued in the clinical arena. The tests are challenging to perform and outside the scope of skills of most hospital microbiology labs. The first descriptions from clinical isolates were usually from endocarditis where bacteria grow slowly. (Antibiotic tolerance among clinical isolates of bacteria. Handwerger S, Tomasz A. Rev Infect Dis. 1985 May-Jun;7(3):368-86.) This report is one of a very few that are beginning to be well investigated and published and thus is a valuable addition to the literature.

The analysis performed here included many techniques which apply to either phenotypic or genotypic tolerance. With so many serial isolates in the study and with examples of both types of tolerance, it would help the reader to reorganize the flow and follow one line of thought for one type of tolerance beginning to end and then go through again for the second (rather than bouncing back and forth over the time line). As stated in the text, phenotypic tolerance is not genetic: rather, the slower the bacteria grow, the slower they die (The rate of killing of *Escherichia coli* by beta-lactam antibiotics is strictly proportional to the rate of bacterial growth. Tuomanen E, Cozens R, Tosch W, Zak O, Tomasz A. J Gen Microbiol. 1986 May;132(5):1297-304.) Importantly, analysis will show that the strain will revert to normal susceptibility when growth rate increases, for instance when biofilm bacteria escape into the rich medium of blood. All the isolates (no need for mutations) in the pacemaker biofilm will be growing slowly and therefore will be phenotypically tolerant, only

some will be genotypically tolerant. This type of analysis could divide the various mutants in this paper into 2 groups and thus simplify the 2 tolerance messages.

Genotypic tolerance can arise by many types of mutations and the main one described here manifested as lengthened lag time. Demonstrating that any one of the specific mutants has a long lag time no matter what medium/site it is in, should be clarified. This would also allow a clear analysis and discussion of the stringent response mutants as they are accepted as affecting tolerance.

Reviewers' comments:

Reviewer #1 (Remarks to the Author):

REVIEW NCOMMS-18-22067

The paper entitled " In-host evolution of *Staphylococcus epidermidis* in a pacemaker associated endocarditis resulting in increased antibiotic tolerance" by Dengler Haunreiter et al. corresponds to a genetic and phenotypic study of *S. epidermidis* isolates originating from a pace maker associated endocarditis.

This type of study analyzing the evolution of isolates originating from the same patient is actually relatively rare and can bring important information on the evolutionary path(s) taken by bacteria when exposed to different stresses within the host. For this reason this work is original and actually represents a first demonstration of a clinically relevant in host evolution of *S. epidermidis*.

In this manuscript the authors remained cautious, may be because this study is performed only with a single patient. This makes it difficult to provide very general conclusions. The authors did actually make one very strong claim. Their main conclusion is indicated in the last 3 lines of the discussion where they wrote that their study confirms the relevance of antibiotic tolerance for chronic and difficult-to-treat bacterial infections. Such demonstration could be very important and studies demonstrating the role of antibiotic tolerance in the failure of classical antibiotic treatment(s) to clear infection are scarce.

We thank the reviewer for the favorable comments and for underlining that '*studies demonstrating the role of antibiotic tolerance in the failure of classical antibiotic treatment(s) to clear infection are scarce*'.

However, with the current version of this manuscript, **I am not totally convinced by the demonstration of the tolerance character of the isolates** (see below). I think that this study also suffers from the fact that no clear conclusions can be drawn on the mutations that are responsible for the observed phenotypes since no mutation reconstruction or reversion to the wild type genotype have been performed.

Major concerns:

- line 206-213: the authors indicated that only one mutation in *rpoB* observed in isolates 16H, K and M was responsible for rifampicin resistance. To me this can be claim only if the mutation is created in a clean (wt) background and if the mutant displays rifampicin resistance. Other unique mutations are found in the isolated 16H, K and M that could be possibly be involved in rifampicin resistance (for example Rsh leu533phe). Additionally isolates 16H, K and M share some mutations with other isolates but it could be that this is the combination of some of this shared mutations with others present in 16H, K and M and absent in others that could be responsible for the rifampicin resistance. Is there any indication in the literature that this specific mutation, if found in other studies, is responsible for rifampicin resistance?

We thank the reviewer for this input. The target of rifampicin is the beta subunit of the DNA-directed RNA polymerase (*RpoB*) and high-level resistance mutants obtained *in vivo* and *in vitro* are due to mutations in *rpoB*, (reviewed in Goldstein). Mutations in single or combined specific positions of the coding sequence are known to affect the drug binding to the RNA polymerase, leading to drug resistance.

The specific insertion we found in the three rifampicin resistant strains, has not been described so far. However, the exact same gene site was found to be mutated (Ala473_Thr) in a resistant *S. aureus* clinical isolate (Wichelhaus *et al.*)

We have expanded the discussion accordingly and have added additional information to both the result and discussion section as well as a new supplementary figure (Figure S3) for clarification.

Goldstein, B. P. Resistance to rifampicin: a review. *J. Antibiot.* **67**, 625–630 (2014)

Wichelhaus, T. A., Schäfer, V., Brade, V. & Böddinghaus, B. Molecular Characterization of *rpoB* Mutations Conferring Cross-Resistance to Rifamycins on Methicillin-Resistant *Staphylococcus aureus*. *Antimicrob Agents Chemother* **43**, 2813–2816 (1999)

- It is difficult to follow the evolution paths leading to the different isolates. To help the reader to follow the evolution of the isolates it will be good to provide a table where for each isolate is indicated all the identified mutations.

We thank the reviewer for this input and we have adapted the Table 2 (Non-synonymous SNPs and InDels found in the clinical ST378 *S. epidermidis*) and the tree in Figure 2B accordingly, which makes it easier to follow for the reader. We have reordered the mutations in Table 2 according to the position in the phylogenetic tree of the clusters/strains they were found in, instead of their position in the genome. As a guide to visualize the mutation characteristics for a specific cluster we have added a last column to the table with symbols. These symbols are included in the corresponding tree branches in Figure 2B.

In addition, it would be very helpful to have a sort of phylogenetic representation of the isolates where the potential overtime emergence of the mutations is indicated, eventually together with the potential associated phenotypes and the sites of isolation of the isolates.

We thank the reviewer for this input. Following the symbol code of Table 2 & Figure 2B, it is now clearer which mutations were found in which cluster/specific strain. The isolation time point of the strains is indicated by the numbering system (i.e. 1A=isolate A from week 1, 14G isolate G from week 14 and 16L isolate L from week 16). In addition, we have highlighted the two phenotypic groups in red and blue throughout the paper and reordered the strains in the figures as they clustered in the tree.

- Why the isolate 16N has not been characterized phenotypically?

We started with a set of 13 strains and only in a second run we sequenced the two remaining strains 16N and 16O. At that time the phenotypic characterization was already completed. We now have performed all phenotypic assays also for 16N and 16O together with the other strains and updated all figures (biofilm, hemolysis, growth curves, doubling times, colony size and MICs) accordingly. The inclusion of more replicates has resulted in the same results with a slight variation as compared to the initially shown data.

- One of the important results of the manuscript is the apparent evolution of the isolates toward increase biofilm formation capacity. Again, it would be good to have a formal demonstration that the detected *agrA* mutations are actually responsible for this phenotype.

We agree with the reviewer that reconstruction of the *agrA* mutations would be the ideal way to directly confirm the effect on biofilm formation. Unfortunately, the clinical *S. epidermidis* isolates were resistant to any state-of-the-art genetic manipulation in our hands. Clinical *S. epidermidis* isolates were previously described to be very difficult to manipulate genetically (eg. Monk *et al.*, Winstel *et al.*).

Therefore, we have added a more detailed literature review on the effect of *agrA* mutations on biofilm formation to the discussion section by referring to Vuong *et al.* and Dai *et al.*

Monk IR, Shah IM, Xu M, Tan M-W, Foster TJ. 2012. Transforming the untransformable: application of direct transformation to manipulate genetically *Staphylococcus aureus* and *Staphylococcus epidermidis*. *mBio* 3(2):e00277-11. doi:10.1128/mBio.00277-11

Winstel V, Kühner P, Krismer B, Peschel A, Rohde H. 2015. Transfer of plasmid DNA to clinical coagulase-negative staphylococcal pathogens by using a unique bacteriophage. *Appl Environ Microbiol* 81:2481–2488. doi:10.1128/AEM.04190-14

Vuong, C., Kocianova, S., Yao, Y., Carmody, A. B. & Otto, M. Increased Colonization of Indwelling Medical Devices by Quorum-Sensing Mutants of *Staphylococcus epidermidis* In Vivo. *J. Infect. Dis.* **190**, 1498–1505 (2004)

Dai, L., Yang L. Parson C., Findlay V. J., Molin S., Qin Z. 2012 *Staphylococcus epidermidis* recovered from indwelling catheters exhibit enhanced biofilm dispersal and “self-renewal” through downregulation of *agr*. *BMC Microbiol.*)

- Concerning the demonstration of the enhanced tolerance of late isolates: in Figure 5C and 5D it would be more accurate to represent the **bacterial survival as log** of the % of bacterial survival. This representation generally reveals more easily important tolerance differences between isolates. No statistics are provided here to support the hypothesis of enhanced tolerance of the late isolates.

We thank the reviewer for pointing out this omission and have added the missing statistics to Figure 5. In addition, as suggested by the reviewer, we have added an alternative visualization of the data with a log scale axis to the supplement (supplementary figure S6A). Since the difference observed mainly happens within a log (100% to 10% at 3 to 6 hours) the percentages give a clearer representation of the data.

Since this is one of the main message of the manuscript it would also be important to potentially show that **what is shown for one early and two late isolates also applied to the different early and late isolates from this study**. Brauner et al in their 2017 manuscript (Brauner, A., Shoshes, N., Fridman, O., & Balaban, N. Q. (2017). An Experimental Framework for Quantifying Bacterial Tolerance. *Biophysical Journal*, 112(12), 2664–2671. <http://doi.org/10.1016/j.bpj.2017.05.014>) proposed an elegant way of measuring/demonstrating enhanced tolerance using the calculation of the **MDK parameter**. The measurement of such as parameter could strengthened the results of this manuscript. One main question that remains unanswered here is what mutation(s) potentially explain the enhanced tolerance of late isolates during the planktonic phase? By using all the isolates in this tolerance assay it could be easier to narrow down common mutations present in all the tolerant isolates. This could eventually for a part replace the necessity to reconstruct identified mutation(s) to validate their association to the tolerance phenotype.

We agree with the reviewer that by analyzing the tolerance phenotype of all isolates one of the main messages is further strengthened. Therefore, we tested the survival of the planktonic cultures of all 15 isolates after 3 hours and after 24 hours ciprofloxacin exposure. This allowed to confirm that group II isolates survive significantly better than group I isolates: while in group I 15.4% bacteria survived the antibiotic challenge, 61.7% survived on average in group II. We added these statistics to the results section and added the Figure 6C.

In group II isolates we found either mutations in the stringent response or an increased lag time (14G, 16L) which could explain the increased antibiotic tolerance compared to group I isolates.

To make the message even clearer, we decided to separate our previous Figure 5 into two figures, leaving the lag time analysis (macro- and microscopic) in Figure 5 and dedicating a new figure, Figure 6, and a corresponding paragraph in the results section to the tolerance phenotype. For clarification, we also added a schematic drawing of the experimental setup to the figure. In the text, we defined the tolerance more quantitatively, making use of the MDK parameter. We show that the isolates differ at the level of the MDK₉₀. The time to kill 90% of the population. i.e. the MDK₉₀, is 3h for 1A and >3h <6h for 14G and 16L.

In Figure 5C the authors showed that biofilms from early or late isolates did not display difference in tolerance. **Does this mean that the mutations causing enhanced tolerance in planktonic do not contribute to tolerance of biofilms?** Since the main lifestyle of bacteria when growing in a prosthetic device is the biofilm lifestyle is it that the enhanced tolerance observed in planktonic situation does actually not contribute to in host tolerance? Why these mutations have been somehow selected if at the end they did not really contributed to the tolerance phenotype in the host? It seems to me that the authors should clarify here their message regarding tolerance in host.

We hypothesize that both the increased lag time and the increased capacity to form biofilm were adaptations relevant under *in-vivo* conditions during the infection. We have expanded the discussion section discussing this hypothesis as well as highlighting the limitations of our assays.

We have also performed additional experiments further underlining that the later isolates 14G and 16L indeed form a biofilm that is less reduced by antibiotic treatment as compared to the biofilm produced by isolate 1A in the *in-vitro* biofilm assay (Figure 6F), even though this is not reflected in the bacterial survival (Figure 6E). Isolate 1A formed a weaker biofilm that contained more live cells before and after antibiotic treatment as compared to the biofilm formed by 14G and 16L. This observation might be explained by the importance of cell lysis as a source for eDNA and so-called moon lighting proteins as investigated in the close relative *S. aureus* (eg. Dengler *et al.*, Foulston *et al.*).

We hypothesize that the robustness of the biofilm was particularly important for the persistence of the biofilm in particular considering the constant flow the pacemaker leads are exposed to.

Further, we hypothesize that the enhanced lag time might have contributed to the manifestation of the bacteremia under antibiotic treatment. Thus, we suggest that: “the bacteremia under antibiotics was most likely a consequence of bacterial seeding from the biofilm and survival of the *S. epidermidis* in the blood due to the observed tolerance phenotype”.

Dengler, V., Foulston, L., Defrancesco, A. S. & Losick, R. An Electrostatic Net Model for the Role of Extracellular DNA in Biofilm Formation by *Staphylococcus aureus*. **197**, 3779–3787 (2015).

Foulston, L., Elsholz, A. K. W., Defrancesco, A. S., Losick, R. & Alicia, S. The Extracellular Matrix of *Staphylococcus aureus* Biofilms Comprises Cytoplasmic Proteins That Associate with the Cell Surface in Response to Decreasing pH. *MBio* **5**, e01667--14 (2014)

Minor concerns:

- one isolate belongs to the ST59 type (isolate 16P). Why no phenotypic analysis has been done on this isolate?

We decided to focus on ST378 isolates for which we had sequential isolates, and which were causing the initial infection. The other two sequence types appeared later during the infection and were isolated only at a single time point. The isolate 16P is a ST59 type and belongs to a ST type unrelated to ST378 (see Tree Figure 2A).

- Isolate 16J from group I follows most of the phenotype of group I with the exception of hemolysis phenotype that resemble group II (figure 3C). Any mutation(s) that could explain this phenotype?

Isolate 16 J contains the following four mutations in addition to the differences between the two groups:

RpoB, DNA-directed RNA polymerase beta subunit	Gln137Pro
Predicted RNA-binding protein, associated with RNase of E/G family	Val15fs
general stress protein 13 (contains ribosomal protein S1 (RPS1) domain)	Thr31fs
Mqo, malate:quinone oxidoreductase	Arg375Trp

We cannot be sure that any of the mutations are responsible for the difference in the hemolysis as none of the genes containing a mutation was previously confirmed to affect hemolysis.

Typo errors:

- line 277: Thus, the variation in in the colony growth ... There is a "in" to remove.

Corrected.

Reviewer #2 (Remarks to the Author):

Antibiotic tolerance is an important clinical problem that is very rarely pursued in the clinical arena. The tests are challenging to perform and outside the scope of skills of most hospital microbiology labs. The first descriptions from clinical isolates were usually from endocarditis where bacteria grow slowly. (Antibiotic tolerance among clinical isolates of bacteria. Handwerker S, Tomasz A. Rev Infect Dis. 1985 May-Jun; 7(3): 368-86.) This report is one of a very few that are beginning to be well investigated and published and thus is a valuable addition to the literature.

The analysis performed here included many techniques which apply to either phenotypic or genotypic tolerance. With so many serial isolates in the study and with examples of both types of tolerance, it would help the reader to **reorganize the flow and follow one line of thought for one type of tolerance beginning to end and then go through again for the second** (rather than bouncing back and forth over the time line).

As stated in the text, phenotypic tolerance is not genetic: rather, the slower the bacteria grow, the slower they die (The rate of killing of Escherichia coli by beta-lactam antibiotics is strictly proportional to the rate of bacterial growth. Tuomanen E, Cozens R, Tosch W, Zak O, Tomasz A. J Gen Microbiol. 1986 May; 132(5): 1297-304.) Importantly, analysis will show that the strain will revert to normal susceptibility when growth rate increases, for instance when biofilm bacteria escape into the rich medium of blood. All the isolates (no need for mutations) in the pacemaker biofilm will be growing slowly and therefore will be phenotypically tolerant, **only some will be genotypically tolerant**. This type of analysis could divide the various mutants in this paper into 2 groups and thus simplify the 2 tolerance messages.

We thank the reviewer for asking for further clarification. We have reorganized the introduction and discussion in order to optimize the flow and to follow one line at a time.

We agree with the reviewer that both genotypic and phenotypic tolerance likely played a role in treatment failure of the patient. We found biofilm on the electrode removed from the patient using SEM. The biofilm was the source of the chronic recurring infection and the break through bacteremia under antibiotics. All the experiments performed in the manuscript were performed using the frozen stock of bacteria which were regrown on agar plates.

Therefore, in the manuscript we focused on the genotypic tolerance, by performing *in vitro* assays to compare the strains that have the exact same genetic background but differ in a few mutations only, since they evolved from a common ancestor strain. We show that adaptation happened very quickly during the infection (high mutation rate) resulting in increased tolerance.

In the paper, all the assays aimed to link the mutations with the phenotypes observed in order to ultimately understand the basis of the genotypic tolerance. Unfortunately, we were not able to point out single mutations. Thus, we formulated possible interpretations/speculations in the discussion but remained cautious.

The last assay, antibiotic clearance of bacteria embedded in a biofilm (and especially the fact that we are comparing this to bacterial killing during the stationary growth phase) is an assay aiming to assess phenotypic tolerance. In order to distinguish clearer between phenotypic and genotypic tolerance, but we made this clearer in the discussion, which comes right after this last result.

Genotypic tolerance can arise by many types of mutations and the main one described here manifested as lengthened lag time. Demonstrating that any one of the specific mutants has a long lag time no matter what medium/site it is in should be clarified. This would also allow a clear analysis and discussion of the stringent response mutants as they are accepted as affecting tolerance.

We thank the reviewer for asking for this clarification. We have performed an additional assay to verify that the relative lag between 1A and 16L was still valid in another medium, i.e. DMEM 4.5g/L glucose +10% FBS. This experiment showed the same trend as observed in TSB: the time for 80% of the population to resumed growth differed by 1.5h between isolate 1A (3.3 ± 0.3 h) and 16L (4.8 ± 0.6 h). We have plotted the results in a new supplementary figure (Figure S8).

We agree that examination of the length of the lag time until growth resumption occurs would have been interesting for all strains. We chose the isolates showing the most extreme growth phenotypes. Accordingly, we have specified the special growth characteristics of the mutants 14G and 16L, which had the highest impact on tolerance. Together with 16O these strains were among the most tolerant. 16O also showed small colony size at 24h and, and a visibly delayed growth curve in liquid medium.

Concerning the other isolates from Group II (namely 16I, 16H, 16K, 16M) they are all indeed stringent response mutants, so probably these mutations had a more direct impact on the tolerance phenotype.

Therefore, it seems that tolerance arose from different sources, mutations in the stringent response as well as an increased lag time.

We integrated these points into our discussion on the genotypic basis of tolerance, which has resulted in a considerably improved discussion.

REVIEWERS' COMMENTS:

Reviewer #1 (Remarks to the Author):

The authors of this study made major efforts to revise their manuscript according to reviewer advices.

I have now only minor remarks.

- The authors acknowledged that they could not performed mutant(s) reconstruction because the collected clinical strains were refractory genetic manipulation. I think this should be indicated in the manuscript so that the readers can understand why it was not possible to reconstruct the identified mutations.
- In the revised version of the manuscript, the authors performed additional phenotypic characterization of the 16N clone. It seems to me that the measured MICs to different antibiotics for this 16N clone are missing in the Supp Table 1. Concerning this clone also the mutation(s) of the *rpoB* gene carried by 16N could be added in Supp Fig. S3.
- in the legend of Figure 2 please indicate the meaning of the star, square, triangle and circle symbols.
- line 322: "we found" has been repeated twice.

Christophe BELOIN

REVIEWERS' COMMENTS:

Reviewer #1 (Remarks to the Author):

The authors of this study made major efforts to revise their manuscript according to reviewer advices.

I have now only minor remarks.

- The authors acknowledged that they could not performed mutant(s) reconstruction because the collected clinical strains were refractory genetic manipulation. I think this should be indicated in the manscript so that the readers can understand why it was not possible to reconstruct the identified mutations.

We thank the reviewer for this input. We added an additional sentence in the Discussion section explaining why we did not perform mutant reconstruction.

- In the revised version of the manuscript, the authors performed additional phenotypic characterization of the 16N clone. It seems to me that the measured MICs to different antibiotics for this 16N clone are missing in the Supp Table 1. Concerning this clone also the mutation(s) of the *rpoB* gene carried by 16N could be added in Supp Fig. S3.

We apologize for the mistake, we uploaded the original version of Supp Table 1 instead of the updated version. Both 16N and 16O MICs are in the updated Supp Table 1.

The mutation of the *rpoB* gene carried by 16N is already shown in Supp Fig S3.

- in the legend of Figure 2 please indicate the meaning of the star, square, triangle and circle symbols.

We now have indicated the meaning of the different symbols in the legend of Figure 2. We also refer to Table 2 for further description of the symbols and the corresponding genetic clusters.

- line 322: "we found" has been repeated twice.

We have corrected this mistake.

Christophe BELOIN